# Improving model calibration with accuracy versus uncertainty optimization

**Ranganath Krishnan**
Intel Labs
ranganath.krishnan@intel.com

**Omesh Tickoo**
Intel Labs
omesh.tickoo@intel.com

## Abstract

Obtaining reliable and accurate quantification of uncertainty estimates from deep neural networks is important in safety-critical applications. A well-calibrated model should be accurate when it is certain about its prediction and indicate high uncertainty when it is likely to be inaccurate. Uncertainty calibration is a challenging problem as there is no ground truth available for uncertainty estimates. We propose an optimization method that leverages the relationship between accuracy and uncertainty as an anchor for uncertainty calibration. We introduce a differentiable *accuracy versus uncertainty calibration* (AvUC) loss function that allows a model to learn to provide well-calibrated uncertainties, in addition to improved accuracy. We also demonstrate the same methodology can be extended to post-hoc uncertainty calibration on pretrained models. We illustrate our approach with mean-field stochastic variational inference and compare with state-of-the-art methods. Extensive experiments demonstrate our approach yields better model calibration than existing methods on large-scale image classification tasks under distributional shift.

## 1   Introduction

Probabilistic deep neural networks (DNNs) enable quantification of principled uncertainty estimates, which are essential to understand the model predictions for reliable decision making in safety critical applications [1]. In addition to obtaining accurate predictions from the model, it is important for the model to indicate when it is likely to make incorrect predictions. Various probabilistic methods have been proposed to capture uncertainty estimates from DNNs including Bayesian [2–8] and non-Bayesian [9, 10] formulations. In spite of recent advances in probabilistic deep learning to improve model robustness, obtaining accurate quantification of uncertainty estimates from DNNs is still an open research problem. A well-calibrated model should be confident about its predictions when it is accurate and indicate high uncertainty when making inaccurate predictions. Modern neural networks are poorly calibrated [11, 12] as they tend to be overconfident on incorrect predictions. Negative log-likelihood (NLL) loss is conventionally used for training the neural networks in multi-class classification tasks. Miscalibration in DNNs has been linked to overfitting of NLL [11, 13]. Probabilistic DNNs fail to provide calibrated uncertainty in between separated regions of observations due to model misspecification and the use of approximate inference [14–16]. Overcoming the problem of poor calibration in modern neural networks is an active area of research [11–21].

In real-world settings, the observed data distribution may shift from training distribution (dataset shift [22]) and there are possibilities of observing novel inputs that are far-off from training data manifold (out-of-distribution). DNN model predictions have been shown to be unreliable under such distributional shift [20, 23, 24]. Obtaining reliable uncertainties even under distributional shift is important to build robust AI systems for successful deployment in the real-world [25, 26]. Uncertainty calibration will also help in detecting distributional shift to caution AI practitioners,

as well-calibrated uncertainty estimates can guide when to trust and when not to trust the model predictions. But uncertainty calibration is a challenging problem due to the unavailability of ground truth uncertainty estimates.

**Contribution**  In this paper, we introduce the *accuracy versus uncertainty calibration* (AvUC) loss function for probabilistic deep neural networks to derive models that will be confident on accurate predictions and indicate higher uncertainty when likely to be inaccurate. We rely on theoretically sound loss-calibrated approximate inference framework [27, 28] with AvUC loss as utilty-dependent penalty term for the task of obtaining well-calibrated uncertainties along with improved accuracy. We find that accounting for predictive uncertainty while training the neural network improves model calibration. To evaluate model calibration under dataset shift, we use various image perturbations and corruptions at different shift intensities [20] and compare with high-performing baselines provided in uncertainty quantification(UQ) benchmark [26]. In summary, we make the following contributions in this work:

- Propose an optimization method that leverages the relationship between accuracy and uncertainty as anchor for uncertainty calibration while training deep neural network classifiers (Bayesian and non-Bayesian). We introduce a differentiable proxy for *Accuracy versus Uncertainty* (AvU) measure and the corresponding *accuracy versus uncertainty calibration* (AvUC) loss function devised to obtain well-calibrated uncertainties, while maintaining or improving model accuracy.
- Investigate accounting for predictive uncertainty estimation in the training objective function and its effect on model calibration under distributional shift (dataset shift and out-of-distribution).
- Propose a post-hoc model calibration method extending the temperature scaling using AvUC loss.
- Empirically evaluate the proposed methods and compare with existing high-performing baselines on large-scale image classification tasks using a wide range of metrics. We demonstrate our method yields state-of-the-art model calibration under distributional shift. We also compare the distributional shift detection performance using predictive uncertainty estimates obtained from different methods.

## 2   Background

**Related work**  Calibration of deep neural networks involves accurately representing predictive probabilities with respect to true likelihood. Existing research to achieve model calibration and robustness in DNNs for multiclass classification tasks can be broadly classified into three categories (i) post-processing calibration (ii) training the model with data augmentation for better representation of training data (iii) probabilistic methods with Bayesian and non-Bayesian formulation for DNNs towards better representation of model parameters. Post-hoc calibration includes temperature scaling [11] and dirichlet calibration [18]. Though post-processing method like temperature scaling perform well under in-distribution conditions, the calibration on the i.i.d. validation dataset does not guarantee calibration under distributional shift [26]. Also they push the accurate predictions to low confidence regions [12]. Data augmentation methods include Mixup [19] and AugMix [21]. Though data augmentation methods improve model robustness, it is practically difficult to introduce a wide spectrum of perturbations and corruptions during training that comprehensively represents the real-world deployment conditions. Deep-ensembles [9] propose a non-Bayesian approach of training an ensemble of neural networks from different random initializations that has been shown to provide calibrated confidence [26]. However, Ensembles introduce additional overhead of training multiple models and significant memory complexity during test time. Approximate Bayesian inference methods for DNNs have been proposed as computing true posterior is intractable, the methods include variational inference [2–4], stochastic gradient variants of MCMC [30, 31], Monte Carlo dropout [5] and SWAG [6]. Approximate Bayesian inference methods are promising, but they may fail to provide calibrated uncertainty in between separated regions of observations as they tend to fit an approximation to a local mode and do not capture the complete true posterior [9, 15, 16, 32]. This may cause the model to be overconfident under distributional shift. Trainable calibration measures [12] have been proposed that encourage confidence calibration during training by optimizing maximum mean calibration error. Snoek et al. [26] show the model calibration degrades with data shift for many of the existing methods that perform well under in-distribution conditions and provides a benchmark evaluating model calibration under data shift. Existing calibration methods do not explicitly account for the quality of predictive uncertainty estimates while training the model, or post-hoc calibration.

**Uncertainty estimation** There are two types of uncertainties that constitute predictive uncertainty of models: *aleatoric uncertainty* and *epistemic uncertainty* [33, 34]. Aleatoric uncertainty captures noise inherent with the observation. Epistemic uncertainty captures the lack of knowledge in representing model parameters. Probabilistic DNNs can quantify both aleatoric and epistemic uncertainties, but deterministic DNNs can capture only aleatoric uncertainty. Various metrics have been proposed to quantify these uncertainties in classification tasks [32, 35], including predictive entropy [36], variation ratio [37] and mutual information [36, 38]. These metrics are rooted with principled foundations in information theory and statistics. In this paper, we use predictive entropy as the uncertainty metric, which represents the predictive uncertainty of the model and captures combination of both epistemic and aleatoric uncertainties [39] in probabilistic models. We will use mean-field stochastic variational inference (SVI) [2, 3] in Bayesian neural networks to illustrate our proposed methods, we refer to Appendix C for background on SVI and uncertainty metrics.

**Loss-calibrated approximate inference** Bayesian decision theory [29] offers a theoretical framework for decision making under uncertainty about a parameter $\theta$. Loss-calibrated approximate inference [27, 28] framework is built upon the basis of Bayesian decision theory to yield optimal predictions for a specific task incorporating a utility function $U(\theta, a)$, which informs us the utility of taking action $a \in \mathcal{A}$. The goal of accomplishing the specific task is defined by the utility function which guides the model learning. Cobb et al. [28] derived a loss-calibrated evidence lower bound comprising of standard evidence lower bound and an additional utility-dependent regularization term.

**Evaluation metrics** We use various metrics[1] to evaluate proposed methods and compare with high-performing Bayesian and non-Bayesian methods under distributional shift. *Expected calibration error* (ECE)↓ [40] is popularly used for determining the calibration of DNNs, which represents the difference in expectation between model accuracy and its confidence. Recently, *expected uncertainty calibration error* (UCE)↓ [41] has been proposed to measure miscalibration of uncertainty, which represents the difference in expectation between model error and its uncertainty. Model calibration is also measured using proper scoring rules [42] such as *negative log likelihood* (NLL)↓ and *Brier score*↓ [43]. The benefits and drawbacks of each of these metrics are described in [26]. Conditional probabilities *p(accurate | certain)*↑ and *p(uncertain | inaccurate)*↑ [39] have been proposed as model performance evaluation metrics for comparing the quality of uncertainty estimates obtained from different probabilistic methods. *p(accurate | certain )* represents the probability that the model is accurate on its output given that it is certain about its predictions, and *p(uncertain | inaccurate )* represents the probability that the model is uncertain about its output given that it has made inaccurate prediction. We also use *area under the receiver operating characteristic curve* (AUROC)↑ [44] and *area under the precision-recall curve* (AUPR)↑ [45] for measuring the distributional shift detection performance, which are typically used for evaluating out-of-distribution detection.

# 3  Obtaining well-calibrated uncertainties with AvUC loss

For evaluating uncertainty estimates from different methods, Mukhoti and Gal [39] had proposed patch accuracy versus patch uncertainty (PAvPU) metric that can be computed per image on semantic segmentation task. Their evaluation methodology was designed based on the assumptions that if a model is confident about its prediction, it should be accurate on the same and if a model is inaccurate on an output, it should be uncertain about the same output.

Extending on these ideas, we leverage the relationship between accuracy and uncertainty as an anchor for uncertainty calibration, since there is no ground-truth for uncertainty estimates. PAvPU metric is not differentiable to be used as a cost function while training the model. We propose differential approximations to the *accuracy versus uncertainty* (AvU) defined in Equation 1 to be used as utility function, which can be computed for a mini-batch of data samples while training the model. We rely on the theoretically sound loss-calibrated approximate inference framework [27, 28] rooted in Bayesian decision theory [29] by introducing AvUC loss as an additional utility-dependent penalty term to accomplish the task of improving uncertainty calibration. A task-specific utility function is employed in Bayesian decision theory to accomplish optimal predictions. In this work, AvU utility function is optimized for the task of obtaining well-calibrated uncertainties (model to provide lower uncertainty for accurate predictions and higher uncertainty towards inaccurate predictions). To estimate the AvU metric during each training step, outputs within a mini-batch can be grouped into four different categories: [i] accurate and certain (AC) [ii] accurate and uncertain (AU) [iii] inaccurate

and certain (IC) [iv] inaccurate an uncertain (IU). $n_{AC}$, $n_{AU}$, $n_{IC}$ and $n_{IU}$ represent the number of samples in the categories AC, AU, IC and IU respectively.

|  |  | Uncertainty | |
|---|---|---|---|
|  |  | certain | uncertain |
| **Accuracy** | accurate | **AC** | **AU** |
|  | inaccurate | **IC** | **IU** |

$$\mathrm{AvU} = \frac{n_{AC} + n_{IU}}{n_{AC} + n_{AU} + n_{IC} + n_{IU}} \tag{1}$$

A reliable and well-calibrated model will provide higher AvU measure ($\mathrm{AvU} \in [0,1]$). Ideally, we expect the model to be certain about its predictions when it is accurate and provide high uncertainty estimates when making inaccurate predictions. We propose differentiable approximations to the AvU utility and introduce a trainable uncertainty calibration loss ($\mathcal{L}_{\mathrm{AvUC}}$) in section 3.1, which serves as the utility-dependent penalty term within the loss-calibrated approximate inference framework described in section 3.2.

## 3.1 Differentiable accuracy versus uncertainty calibration (AvUC) loss

**Notations** Consider a multi-class classification problem on a large labeled dataset with N examples and K classes denoted by $\mathcal{D} = \{(x_n, y_n)\}_{n=1}^{N}$. Dataset is partitioned into M mini-batches i.e. $\mathcal{D} = \{\mathcal{D}_m\}_{m=1}^{M}$, each containing B=N/M examples. During training, we process a group of randomly sampled examples (mini-batch) $\mathcal{D}_m = \{(x_i, y_i)\}_{i=1}^{B}$ per iteration. For each example with input $x_i \in \mathcal{X}$ and $y_i \in \mathcal{Y} = \{1, 2, \cdots, k\}$ representing the ground-truth class label, let $p_i(y|x_i, w)$ be the output from the neural network $f_w(y|x_i)$. In case of probabilistic models, predictive distribution is obtained from T stochastic forward passes (Monte Carlo samples), $p_i(y|x_i, w) = \frac{1}{T} \sum_{t=1}^{T} p_i^t(y|x_i, w_t)$. Let us define $\hat{y}_i = \arg\max_{y \in \mathcal{Y}} p_i(y|x_i, w)$ as the predicted class label, $p_i = \max_{y \in \mathcal{Y}} p_i(y|x_i, w)$ and $u_i = -\sum_{y \in \mathcal{Y}} p_i(y|x_i, w) \log p_i(y|x_i, w)$ as confidence (probability of predicted class) and predictive uncertainty estimate for the model prediction respectively. $u_{th}$ is the threshold above which prediction is considered to be uncertain, and $\mathbb{1}$ is the indicator function.

$$n_{AU} := \sum_i \mathbb{1}(\hat{y}_i = y_i \ and \ u_i > u_{\mathrm{th}}) \quad ; \quad n_{IC} := \sum_i \mathbb{1}(\hat{y}_i \neq y_i \ and \ u_i \leq u_{\mathrm{th}})$$

$$n_{AC} := \sum_i \mathbb{1}(\hat{y}_i = y_i \ and \ u_i \leq u_{\mathrm{th}}) \quad ; \quad n_{IU} := \sum_i \mathbb{1}(\hat{y}_i \neq y_i \ and \ u_i > u_{\mathrm{th}}) \tag{2}$$

We define the AvUC loss function representing negative log AvU in Equation 3. In order to make the loss function differentiable with respect to the neural network parameters, we define proxy functions to approximate $n_{AC}$, $n_{AU}$, $n_{IC}$ and $n_{IU}$ as given by Equations 4. The hyperbolic tangent function is used to scale the uncertainty values between 0 and 1, $\tanh(u_i) \in [0,1]$. The intuition behind these approximations is that the probability of the predicted class $\{p_i \to 1\}$ when the predictions are accurate and $\{p_i \to 0\}$ when inaccurate. Also, the scaled uncertainty $\{\tanh(u_i) \to 0\}$ when the predictions are certain and $\{\tanh(u_i) \to 1\}$ when uncertain. Under ideal conditions, these proxy functions in Equation 4 will be equivalent to indicator functions defined in Equations 2. This loss function can be used with standard gradient descent optimization and enables the model to learn to provide well-calibrated uncertainties, in addition to improved prediction accuracy. Minimizing the AvUC loss function is equivalent to maximizing AvU measure (Equation 1). The AvUC loss will be perfect 0 only when all the accurate predictions are certain and inaccurate predictions are uncertain.

$$\mathcal{L}_{\mathrm{AvUC}} := -\log\left(\frac{n_{AC} + n_{IU}}{n_{AC} + n_{IU} + n_{AU} + n_{IC}}\right) = \log\left(1 + \frac{n_{AU} + n_{IC}}{n_{AC} + n_{IU}}\right) \tag{3}$$

where;

$$n_{AU} = \sum_{i \in \left\{\begin{smallmatrix}\hat{y}_i = y_i \ and \\ u_i > u_{\mathrm{th}}\end{smallmatrix}\right\}} p_i \odot \tanh(u_i) \quad ; \quad n_{IC} = \sum_{i \in \left\{\begin{smallmatrix}\hat{y}_i \neq y_i \ and \\ u_i \leq u_{\mathrm{th}}\end{smallmatrix}\right\}} (1 - p_i) \odot (1 - \tanh(u_i))$$

$$n_{AC} = \sum_{i \in \left\{\begin{smallmatrix}\hat{y}_i = y_i \ and \\ u_i \leq u_{\mathrm{th}}\end{smallmatrix}\right\}} p_i \odot (1 - \tanh(u_i)) \quad ; \quad n_{IU} = \sum_{i \in \left\{\begin{smallmatrix}\hat{y}_i \neq y_i \ and \\ u_i > u_{\mathrm{th}}\end{smallmatrix}\right\}} (1 - p_i) \odot \tanh(u_i)$$

$$\tag{4}$$

AvUC loss is devised to improve uncertainty calibration that can be used as an additional penalty term and combined with existing losses without modifying the underlying principles (e.g. ELBO for Bayesian DNN, cross-entropy for non-Bayesian DNN classifier). AvUC enables uncertainty calibration by overcoming the challenge of unavailability of ground truth uncertainty estimates while accounting for the quality of principled aleatoric and epistemic uncertainties, which are important for many applications.

## 3.2 Loss-calibrated approximate inference with AvUC loss

The loss-calibrated evidence lower bound (ELBO) is defined in Equation 5 that incorporates AvUC loss as an additional utility-dependent penalty term and $\beta$ is the hyperparameter for relative weighting of AvUC loss with respect to ELBO. We illustrate our method with mean-field stochastic variational inference (SVI) [2, 3]. Our implementation is shown in Algorithm 1 and we refer to this method as **SVI-AvUC**. The operations 16-19 in Algorithm 1 are the additional steps with respect to standard SVI.

$$\mathcal{L} := \underbrace{\underbrace{-\mathbb{E}_{q_\theta(\mathrm{w})}[\log p(\mathrm{y}|\mathrm{x}, \mathrm{w})]}_{\text{expected negative log likelihood}} + \underbrace{\mathrm{KL}[q_\theta(\mathrm{w})||p(\mathrm{w})]}_{\text{Kullback-Leibler divergence}}}_{\mathcal{L}_{\text{ELBO}} (\text{negative ELBO})} + \beta \underbrace{\log\left(1 + \frac{\mathrm{n}_{AU} + \mathrm{n}_{IC}}{\mathrm{n}_{AC} + \mathrm{n}_{IU}}\right)}_{\mathcal{L}_{\text{AvUC}}(\text{AvUC loss})} \quad (5)$$

---

**Algorithm 1** SVI-AvUC optimization

---

1: Given dataset $D = \{X, Y\}$
2: let variational parameters $\theta = (\mu, \rho)$      ▷ approx variational posterior $q_\theta(\mathrm{w}) = \mathcal{N}(\mu, \log(1 + e^\rho))$
3: set the weight priors, $p(\mathrm{w}) := \mathcal{N}(0, I)$
4: initialize $\mu$ and $\rho$
5: define learning rate schedule $\alpha$
6: **repeat**
7:     Sample $B$ index set of training samples; $\mathcal{D}_m = \{(\mathrm{x}_i, \mathrm{y}_i)\}_{i=1}^{B}$       ▷ batch-size
8:     **for** $i \in B$ **do**
9:        **for** $t \leftarrow 1$ to $T$ **do**           ▷ T Monte Carlo samples
10:           Sample $\epsilon \sim \mathcal{N}(0, I)$
11:           $\mathrm{w}_t = \mu + \log(1 + \exp(\rho)) \odot \epsilon$        ▷ $\odot$ represents pointwise multiplication
12:           $p_i^t(\mathrm{y}|\mathrm{x}_i, \mathrm{w}_t) = f_{\mathrm{w} \sim q_\theta(\mathrm{w})}(\mathrm{y}|\mathrm{x}_i)$    ▷ perform a stochastic forward pass with sampled weight
13:        **end for**
14:        Obtain predictive distribution from $T$ stochastic forward passes
          $p_i(\mathrm{y}|\mathrm{x}_i, \mathrm{w}) = \frac{1}{T} \sum_{t=1}^{T} p_i^t(\mathrm{y}|\mathrm{x}_i, \mathrm{w}_t)$
15:        Obtain predicted label and probability of predicted class
          $\widehat{y}_i \leftarrow \underset{\mathrm{y} \in \mathrm{Y}}{\mathrm{argmax}}\, p_i(\mathrm{y}|\mathrm{x}_i, \mathrm{w})$     ;     $p_i \leftarrow \underset{\mathrm{y} \in \mathrm{Y}}{\max}\, p_i(\mathrm{y}|\mathrm{x}_i, \mathrm{w})$
16:        Calculate predictive uncertainty                ▷ predictive entropy
          $\mathbf{u}_i = -\sum_k \left(\frac{1}{T} \sum_t p_i^t(\mathrm{y} = k|\mathrm{x}_i, \mathrm{w}_t)\right) \log\left(\frac{1}{T} \sum_t p_i^t(\mathrm{y} = k|\mathrm{x}_i, \mathrm{w}_t)\right)$    ▷ where; $\mathrm{w}_t \sim q_\theta(\mathrm{w})$
17:     **end for**
18:     Compute $\mathbf{n_{AC}}, \mathbf{n_{AU}}, \mathbf{n_{IC}}, \mathbf{n_{IU}}$             ▷ Equations 4
19:     Compute loss-calibrated ELBO (total loss), $\mathcal{L} = \mathcal{L}_{\text{ELBO}} + \mathcal{L}_{\textbf{AvUC}}$      ▷ Equation 5
20:     Compute the gradients of loss function w.r.t $\mu$ and $\rho$, $\Delta\mathcal{L}_\mu$ and $\Delta\mathcal{L}_\rho$ respectively
21:     Update the variational parameters $\mu$ and $\rho$
       $\mu \leftarrow \mu - \alpha\Delta\mathcal{L}_\mu$
       $\rho \leftarrow \rho - \alpha\Delta\mathcal{L}_\rho$
22: **until** $\mu$ and $\rho$ has converged, or when stopped

---

AvU is the utility function which guides optimal predictions in accomplishing the task of getting well-calibrated uncertainties and proposed AvUC loss serves as an utility-dependent penalty term within the loss-calibrated inference framework. For the initial few epochs, we train the model only with ELBO loss as this allows to learn the uncertainty threshold required for AvUC loss[2].The threshold is obtained from the average of predictive uncertainty mean for accurate and inaccurate predictions on the training data from initial epochs.

Theoretically AvUC loss will be equal to 0 only when the model's uncertainty is perfectly calibrated (utility function is maximized, AvU=1). As noted in Equations 3 and 4, AvUC loss attempts to

maximize the utility function AvU, which will indirectly push the values of uncertainties up or down based on the accuracy of predictions. When uncertainty estimates are not accurate, $\text{AvU} \to 0$ and $\mathcal{L}_{\text{AvUC}} \to \infty$ guiding the gradient computation exert AvUC loss towards 0, which will happen when AvU score is pushed higher ($\text{AvU} \to 1$), enabling the model to maximize the utility to provide well-calibrated uncertainties. In Appendix D.1, we show how AvUC loss and ELBO loss vary during training and the impact of AvUC regularization term on loss-calibrated ELBO (total loss) and actual AvU score.

## 3.3  Post-hoc model calibration with AvU temperature scaling (AvUTS)

We propose post-hoc uncertainty calibration for pretrained models by extending the temperature scaling [11] methodology to optimize the AvUC loss instead of NLL. The optimal temperature $\text{T} > 0$, a scalar value to rescale the logits of final layer is identified by minimizing the AvUC loss as defined in Equation 3 on held-out validation set. The uncertainty threshold [2] required for calculating $n_{AC}, n_{AU}$, $n_{IC}$ and $n_{IU}$ is obtained by finding the average predictive uncertainty for accurate and inaccurate predictions from the uncalibrated model using the same held-out validation data $\mathcal{D}_{\text{v}} = \{(\text{x}_\text{v}, \text{y}_\text{v})\}_{\text{v}=1}^{V}$, $\text{u}_{th} = \left( \frac{\overline{\text{u}}_{(\hat{y}_\text{v} = y_\text{v})} + \overline{\text{u}}_{(\hat{y}_\text{v} \neq y_\text{v})}}{2} \right)$. We refer this method applied to pretrained SVI model as **SVI-AvUTS**.

## 4  Experiments and Results

We perform a thorough empirical evaluation of our proposed methods SVI-AvUC and SVI-AvUTS on large-scale image classification task under distributional shift. We evaluate the model calibration; model performance with respect to confidence and uncertainty estimates; and the distributional shift detection performance. We use ResNet-50 and ResNet-20 [46] DNN architectures on ImageNet [47] and CIFAR10 [48] datasets respectively. We compare the proposed methods with various high performing non-Bayesian and Bayesian methods including vanilla DNN (Vanilla), Temperature scaling (Temp scaling) [11], Deep-ensembles (Ensemble) [9], Monte Carlo dropout (Dropout) [5], Mean-field stochastic variational inference (SVI) [2, 3], Temperature scaling on SVI (SVI-TS) and Radial Bayesian neural network (Radial BNN) [8]. In Appendix D, we compare with additional methods, Dropout and SVI on the last layer of neural network (LL-Dropout and LL-SVI) [49, 50]. The work from Snoek et al. [26] suggests SVI is very promising on small-scale problems, but is difficult to scale to larger datasets. We choose SVI as a baseline to illustrate our methods AvUC and AvUTS. We were able to scale SVI to the large-scale ImageNet dataset with ResNet-50 by specifying the weight priors and initializing the variational parameters using Empirical Bayes method following [51]. The results for the methods: Vanilla, Temp scaling, Ensemble, Dropout, LL Dropout and LL SVI are obtained from the model predictions provided in UQ benchmark [26] and we follow the same methodology for model evaluation under distributional shift by utilizing 16 different types of image corruptions at 5 different levels of intensities for each datashift type proposed in [20], resulting in 80 variations of test data for datashift evaluation. We refer to Appendix A for details on datashift types used in experiments, along with visual examples. All the methods are compared both under in-distribution and distributional shift conditions with same evaluation criteria for fair comparison. For SVI-AvUC implementation, we use the same hyperparameters as SVI baseline. We provide details of our model implementations and hyperparameters for SVI, SVI-TS, SVI-AvUC, SVI-AvUTS and Radial BNN in Appendix B.

**Model calibration evaluation** We evaluate model calibration under in-distribution and dataset shift conditions following methodology in [26]. Figure 1 shows the comparison of ECE, UCE and accuracy from different methods for test data (in-distribution) and dataset shift summarizing across 80 variations of shifted data on both ImageNet and CIFAR10. ECE represents the model calibration error with respect to confidence (probability of predicted class) and UCE represents the model calibration error with respect to predictive uncertainty representing entire predictive distribution of probabilities across the classes. A reliable and well-calibrated model should provide low calibration errors even with increased intensity of data shift, though accuracy may degrade with data shift. From Figure 1, we can see the model accuracy reduces with increased data shift intensity and Ensembles method provides highest accuracy among existing methods. With model calibration, post-hoc calibration method SVI-AvUTS improves results over SVI baseline and SVI-AvUC outperforms all the methods by providing lower calibration errors (both ECE and UCE) at increased data shift levels while providing comparable model accuracy to Ensembles. We provide additional results (NLL, Brier score) and tables with numerical data comparison in Appendix D.2.

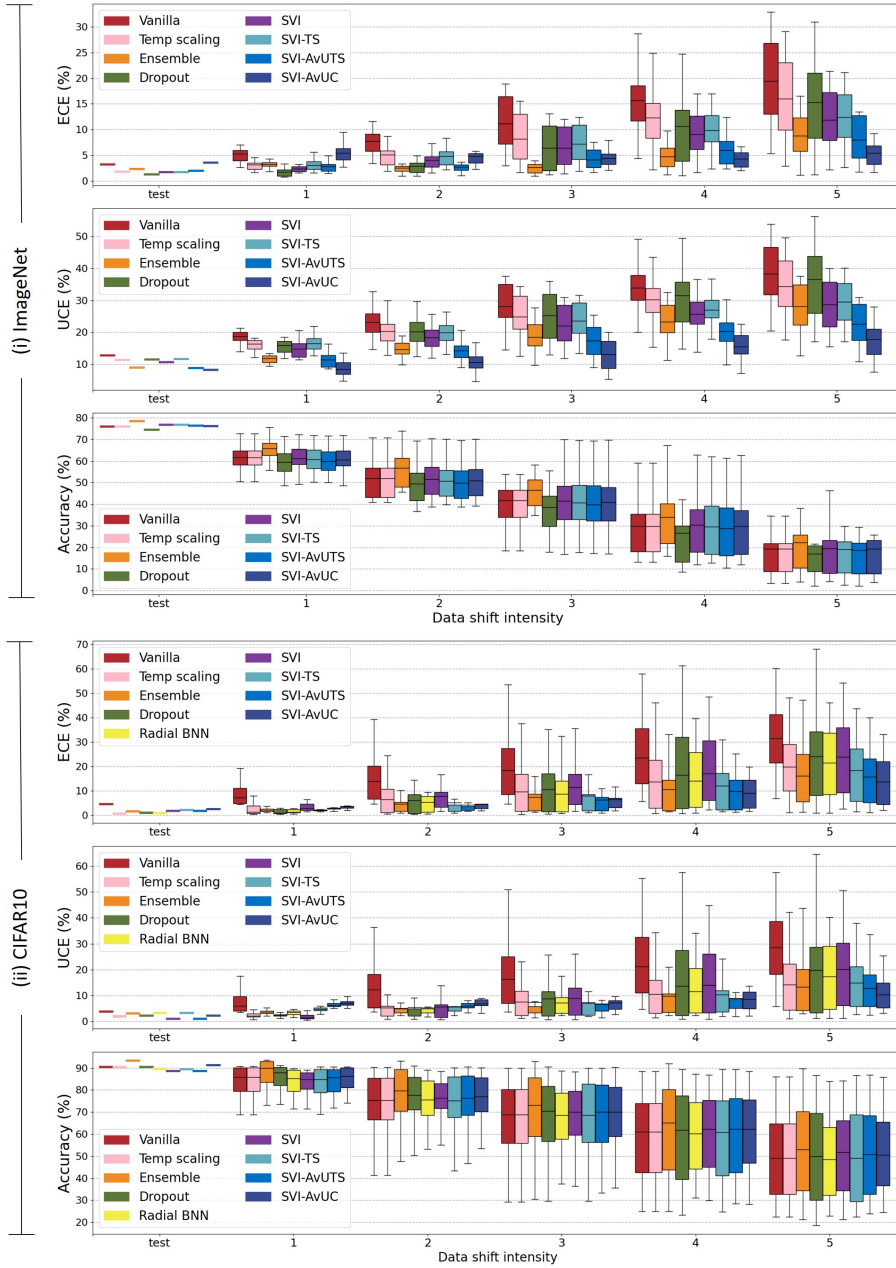

Figure 1: Model calibration comparison using ECE↓ and UCE↓ on (i) ImageNet and (ii) CIFAR10 under in-distribution (test) and dataset shift at different levels of shift intensities (1-5). A well-calibrated model should provide lower calibration errors even at increased datashift, though accuracy may degrade. At each shift intensity level, the boxplot summarizes the results across 16 different datashift types showing the min, max, mean and quartiles. **SVI-AvUC** provides lower ECE and UCE at increased dataset shift demonstrating it yields better model calibration compared to other methods. Spearman rank-order correlation coefficients [55] assessing the relationship between calibration errors and dataset shift is provided in Appendix D.2.

Table 1: Additional results evaluating AvUC and AvUTS methods applied to Vanilla baseline on CIFAR10. Vanilla-AvUTS and Vanilla-AvUC provides lower ECE and UCE (mean across 16 different data shift types) compared to the baseline.

| Method | ECE (%)↓ at various datashift intensities | | | | | | UCE (%)↓ at various datashift intensities | | | | | |
|---|---|---|---|---|---|---|---|---|---|---|---|---|
| | 0 | 1 | 2 | 3 | 4 | 5 | 0 | 1 | 2 | 3 | 4 | 5 |
| Vanilla | 4.6 | 9.8 | 13.9 | 18.3 | 23.6 | 31.5 | 3.8 | 8.5 | 12.2 | 16.2 | 21.2 | 28.5 |
| Vanilla-AvuTS | **2.1** | **4.3** | **7.3** | 11.8 | 15.0 | 27.7 | **1.1** | **2.6** | 4.7 | 8.6 | 11.7 | 23.1 |
| Vanilla-AvUC | 3.4 | 5.7 | 8.4 | **11.4** | **14.3** | **23.2** | 1.7 | 2.8 | **4.6** | **6.9** | **9.3** | **16.8** |

In addition to SVI-AvUC and SVI-AvUTS, we evaluate AvUC and AvUTS methods applied to vanilla baseline with entropy of softmax used as the predictive uncertainty in computing AvUC loss, which is combined with the cross-entropy loss. Table 1 shows AvUTS and AvUC improves the model calibration errors (ECE and UCE) on the vanilla baseline as well.

**Model confidence and uncertainty evaluation** We evaluate the quality of confidence measures using *accuracy vs confidence* plots following the methodology from [9, 26]. We evaluate the quality of predictive uncertainty estimates with *p(uncertain | inaccurate)* and *p(accurate | certain)* metrics across various uncertainty thresholds as proposed in [39]. A reliable model should be accurate when it is certain about its prediction and indicate high uncertainty when it is likely to be inaccurate. Figures 2(a) and (b) show SVI-AvUC is more accurate at higher confidence, Figure 2(c) show SVI-AVUC is more accurate at lower uncertainty (being certain). Figures 2(d), (e), (f) shows SVI-AvUC is more uncertain when making inaccurate predictions under distributional shift, compared to other methods. Figures 2(g) and (h) show SVI-AvUC has lesser number of examples with higher confidence when model accuracy is low under distributional shift. Figure 2(i) show SVI-AvUC provides higher predictive entropy on out-of-distribution data. We provide additional results in Appendix D.3. In summary, SVI-AvUTS improves the quality of confidence and uncertainty measures over the SVI baseline, while preserving or improving accuracy. SVI-AvUC outperforms other methods in providing calibrated confidence and uncertainty measures under distributional shift.

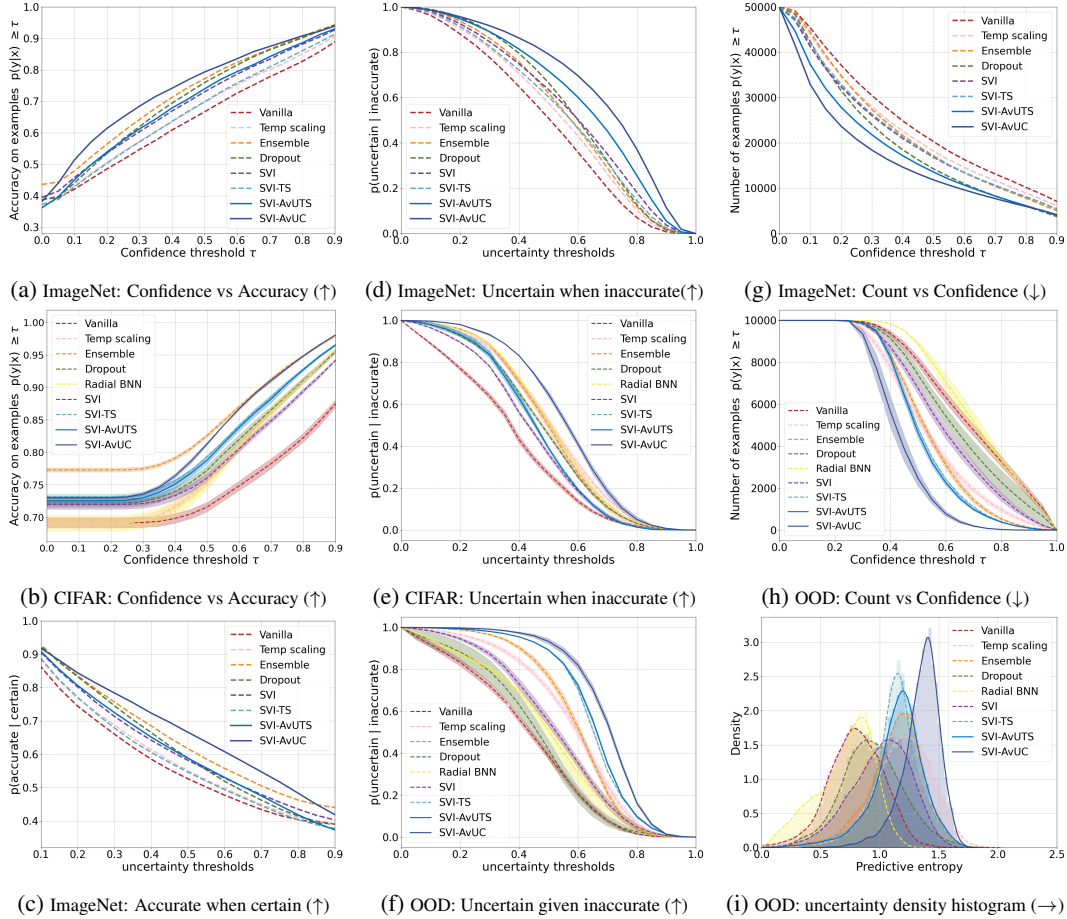

Figure 2: Model confidence and uncertainty evaluation under distributional shift (dataset shift on ImageNet and CIFAR10 with Gaussian blur of intensity 3, SVHN [52] is used as out-of-distribution (OOD) on model trained with CIFAR10). Column 1: 2(a) and (b) show accuracy as a function of confidence, 2(c) show probability of model being accurate when certain about its predictions. Column 2: 2(d), (e) and (f) show probability of model being uncertain when making inaccurate predictions. Normalized uncertainty thresholds $t \in [0, 1]$ are shown in plots as the uncertainty range varies for different methods. Column 3: 2(g) and (h) show number of examples above given confidence value and 2(i) shows density histogram of entropy on OOD data.

**Distributional shift detection**   We evaluate the performance of detecting distributional shift in neural networks using uncertainty estimates. This is a binary classification problem of identifying if an input sample is from in-distribution or shifted data. We evaluate using *AUROC*, *AUPR* and *detection accuracy* metrics following the methodology in [10]. We expect higher uncertainty under distributional shift as model tends to make inaccurate predictions and lower uncertainty for in-distribution data. In Figure 3, we see better separation of predictive uncertainty densities for SVI-AvUC as compared to other methods, which is also quantified with Wasserstein distance [53]. In Table 2, we present the dataset shift detection performance for ImageNet and CIFAR10 shifted with Gaussian blur at intensity 5. We also provide the out-of-distribution detection performance when the model trained with CIFAR10 is introduced with SVHN data during test time. Results in Table 2 show SVI-AvUC outperforms other methods in distributional shift detection.

| Method | Vanilla | Temp scaling | Ensemble | Dropout | SVI | SVI-TS | SVI-AvUTS | SVI-AvUC |
|---|---|---|---|---|---|---|---|---|
| Wasserstein distance | 2.7319 | 2.9098 | 3.0219 | 3.2837 | 3.7311 | 3.6961 | 3.9443 | **4.2887** |

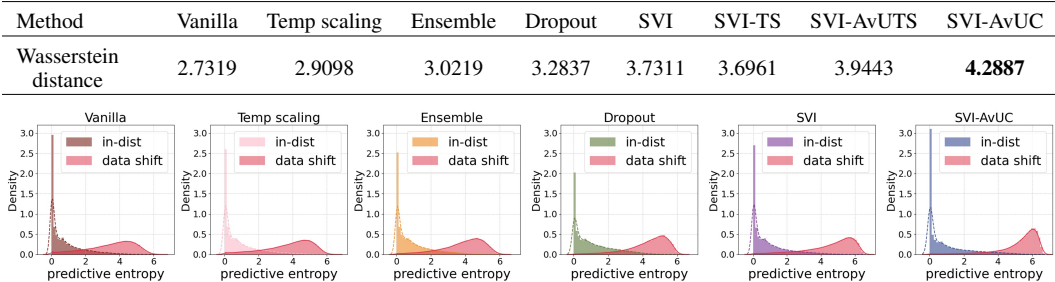

Figure 3: Density histograms of predictive uncertainty estimates on ImageNet in-distribution test set and data shifted with Gaussian blur of intensity 5. SVI-AvUC shows best separation of densities between in-distribution and data-shift as quantified by Wasserstein distance.

Table 2: Distributional shift detection using predictive uncertainty. For dataset shift detection on ImageNet and CIFAR10, we use test data shifted with Gaussian blur of intensity 5. SVHN is used as out-of-distribution(OOD) data for OOD detection on model trained with CIFAR10. All values are in percentages and best results are indicated in bold. SVI-AvUC outperforms across all the metrics.

| Method | ImageNet (Dataset shift detection) | | | | CIFAR10 (Dataset shift detection) | | | | CIFAR10 (OOD detection) | | | |
|---|---|---|---|---|---|---|---|---|---|---|---|---|
| | AUROC ↑ | Detection accuracy↑ | AUPR in↑ | AUPR out↑ | AUROC ↑ | Detection accuracy↑ | AUPR in↑ | AUPR out↑ | AUROC ↑ | Detection accuracy↑ | AUPR in↑ | AUPR out↑ |
| Vanilla DNN [46] | 93.36 | 86.08 | 92.82 | 93.71 | 92.36 | 85.78 | 93.81 | 89.87 | 96.53 | 91.60 | 97.23 | 95.23 |
| Temp scaling [11] | 93.71 | 86.47 | 93.21 | 94.01 | 92.71 | 86.72 | 94.21 | 90.11 | 96.65 | 92.14 | 97.39 | 95.29 |
| Ensemble [9] | 95.49 | 88.82 | 95.31 | 95.64 | 90.71 | 83.94 | 92.55 | 87.68 | 95.78 | 91.47 | 96.95 | 92.65 |
| Dropout [5] | 96.38 | 89.98 | 96.16 | 96.67 | 87.64 | 81.20 | 89.83 | 83.13 | 91.48 | 86.84 | 93.99 | 86.37 |
| SVI [3] | 96.40 | 90.03 | 95.97 | 96.83 | 85.89 | 79.31 | 88.34 | 81.48 | 93.94 | 87.87 | 95.30 | 91.61 |
| SVI-TS [3, 11] | 96.61 | 90.45 | 96.24 | 96.98 | 81.08 | 75.43 | 84.85 | 74.16 | 90.81 | 87.59 | 93.84 | 82.18 |
| SVI-AvUTS | 96.89 | 90.93 | 96.58 | 97.19 | 81.19 | 75.82 | 85.09 | 74.17 | 93.79 | 89.39 | 95.49 | 87.99 |
| **SVI-AvUC** | **97.60** | **92.07** | **97.39** | **97.85** | **95.54** | **88.43** | **96.32** | **94.66** | **99.35** | **97.16** | **99.50** | **98.91** |

## 5   Conclusion

We introduced the *accuracy versus uncertainty calibration* (AvUC) loss and proposed novel optimization methods AvUC and AvUTS for improving uncertainty calibration in deep neural networks. Uncertainty calibration is important for reliable and informed decision making in safety critical applications, we envision AvUC as a step towards advancing probabilistic deep neural networks in providing well-calibrated uncertainties along with improved accuracy. Our work shows that accounting for uncertainty estimation during training can improve model calibration significantly. We demonstrated our method SVI-AvUC provides better model calibration than existing state-of-the-art methods under distributional shift. We showed our simple post-hoc calibration method AvUTS can improve the uncertainty calibration over the baseline. We also demonstrated the effectiveness of proposed methods in detecting distributional shift while outperforming the other methods. We have illustrated AvUC and AvUTS on stochastic variational inference (Bayesian) and vanilla (non-Bayesian) methods. We have made the code [3] available to facilitate probabilistic deep learning community to evaluate and improve model calibration for various other baselines.

## Broader Impact

As AI systems backed by deep learning are used in safety-critical applications like autonomous vehicles, medical diagnosis, robotics etc., it is important for these systems to be explainable and trustworthy for successful deployment in real-world. Having the ability to derive uncertainty estimates provides a big step towards explainability of AI systems based on Deep Learning. Having calibrated uncertainty quantification provides grounded means for uncertainty measurement in such models. A principled way to measure reliable uncertainty is the basis on which trustworthy AI systems can be built. Research results and multiple resulting frameworks have been released for AI Fairness measurement that base components of fairness quantification on uncertainty measurements of classified output of deep learning models. We believe that our work can be a big step towards measuring such uncertainties in a reliable fashion. The resulting, well calibrated, uncertainty measures can then be used as an input for building fair and trustworthy AI models that implement explainable behavior. This explanation is also critical for building AI systems that are robust to adversarial blackbox and whitebox attacks. These well calibrated uncertainties can guide AI practitioners to better understand the predictions for reliable decision making, i.e. to know "when to trust" and "when not to trust" the model predictions (especially in high-risk domains like healthcare, financial, legal etc). In addition, calibrated uncertainty opens the doors for wider adoption of deep network architectures in interesting applications like multimodal fusion, anomaly detection and active learning. Using calibrated uncertainty as a measure for distributional shift (out-of-distribution and dataset shift) detection is also a key enabler for self-learning systems that form a critical component of realizing the dream of Artificial General Intelligence (AGI).

## Acknowledgement

We would like to thank Mahesh Subedar (Intel Labs), Willem M Beltman (Intel Labs) and the reviewers for their comments and feedback that helped to improve the manuscript.

## Footnotes

[1]Arrows next to each evaluation metric indicate which direction is better. Equations in Appendix C.3

[2]We also optimized area under the curve of AvU across various uncertainty thresholds towards a threshold free mechanism as presented in Appendix F, but the results are similar except being more compute intensive during training.

[3] https://github.com/IntelLabs/AVUC

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
