[Supplementary Material]

# Appendix: Improving model calibration with accuracy versus uncertainty optimization

## A  Dataset shift

We use various image corruptions and perturbations proposed by Hendrycks and Dietterich [20] for evaluating model calibration under dataset shift, following the methodology in uncertainty quantification (UQ) benchmark [26]. We evaluate our proposed methods with the high performing baselines provided in the UQ benchmark. For dataset shift evaluation, 16 different types of image corruptions at 5 different levels of intensities are utilized, resulting in 80 variants of datashift. Figure F1 shows an example of 16 different datashift types on ImageNet used in our experiments during test time. Figure F2 shows an example of different shift intensities (from level 1 to 5) for Gaussian blur. The same datashifts are applied to CIFAR10 as well. These dataset shifts are encountered during test time only, the models are trained with clean data.

Figure F1: Example of sixteen different image corruptions [20] used during test time (dataset shift)

Figure F2: Example of Gaussian blur at different levels of shift intensity (1-5)

## B  Experimental details and Parameters

**Codebase**  We have made our code available open-source at `https://github.com/intelLabs/AVUC`. We have implemented the code necessary for our experiments of SVI (*mean-field stochastic variational inference*), SVI-AvUC (*accuracy vs uncertainty calibration*) and SVI-AvUTS (*accuracy vs uncertainty temperature scaling*) in PyTorch [54], including AvUC loss and variational layers support required for stochastic variational inference.

### B.1  Model details

In this section we describe all hyper-parameters used for training the models and evaluation we performed in Section 4. On CIFAR10 and ImageNet image classification tasks under distributional shift, we use ResNet-20 and ResNet-50 [46] architectures respectively. The results for the methods: Vanilla, Temp scaling, Ensemble, Dropout, LL Dropout and LL SVI are computed from the model predictions provided in UQ benchmark [26].

#### B.1.1  CIFAR10/ResNet-20

**SVI-AvUC**  We use the same hyper-parameters as Snoek et al. [26] used for SVI on CIFAR10 for fair comparison. The models were trained with Adam optimizer for 200 epochs with initial learning

rate of $1.189e^{-3}$ and batch size of $107$. As part of the learning rate schedule, initial learning rate was multiplied by 0.1, 0.01, 0.001 and 0.0005 at epochs 80, 120, 160 and 180 respectively. The training samples were distorted with random horizontal flips and random crops with 4-pixel padding as mentioned in [46]. We used $\beta = 3$ in Equation 5 for relative weighting of AvUC loss with respect to ELBO loss. We used $128$ Monte Carlo samples from weight posterior for evaluation.

**SVI-AvUTS**  We find the optimal temperature for pretrained SVI model by minimizing the *accuracy versus uncertainty calibration* (AvUC) loss on hold-out validation data. We adapted the code from [11] and replaced negative log-likelihood loss with our AvUC loss implementation for optimization at learning rate of 0.005. The CIFAR10 training data was split into 9:1 ratio (45k train set and 5k hold-out validation set images). The SVI baseline model was trained with same hyper-parameters as in UQ benchmark [26], described above.

**Radial BNN**  To compare our methods SVI-AvUC and SVI-AvUTS with Radial BNN, we implemented ResNet-20 for Radial BNN adapting the code from [8]. The models were trained with Adam optimizer for 200 epochs with initial learning rate of $1e^{-3}$ and batch size of $256$. As part of the learning rate schedule, initial learning rate was multiplied by 0.1, 0.01, 0.001 and 0.0005 at epochs 80, 120, 160 and 180 respectively. The training samples were distorted with random horizontal flips and random crops with 4-pixel padding as mentioned in [46]

We evaluate with 10k test images, along with 80 variants of dataset shift (each with 10k images) that includes 16 different types of datashift at 5 different intensities as described in Section A.

For out-of-distribution (OOD) evaluation, we use SVHN dataset as OOD data on models trained with CIFAR10.

### B.1.2 ImageNet/ResNet-50

**SVI**  In order to scale SVI to large-scale ImageNet dataset and ResNet-50 model, we specify the weight priors and initialize the variational parameters using Empirical Bayes method as proposed in [51]. The weights are modeled with fully factorized Gaussian distributions represented by $\mu$ and $\sigma$. In order to ensure non-negative variance, $\sigma$ is expressed in terms of softplus function with unconstrained parameter $\rho$, i.e. $\sigma = \log(1 + exp(\rho))$. The weight prior is set to $\mathcal{N}(\mathrm{w}_{MLE}, I)$ and the variational parameters $\mu$ and $\rho$ are initialized with $\mathrm{w}_{MLE}$ and $\log(e^{\delta|\mathrm{w}_{MLE}|} - 1)$ repectively. The initial maximum likelihood estimate (MLE) for weights $\mathrm{w}_{MLE}$ are obtained from pretained ResNet-50 model available in the torchvision package[1] and $\delta$ is set to 0.5. The model was trained for 50 epochs using SGD optimizer with initial learning rate of $0.001$, momentum of 0.9, weight decay of $1e^{-4}$ and batch size of $96$. We used learning rate schedule that multiplies the learning rate by 0.1 every 30 epochs.The training samples were distorted with random horizontal flips and random crops as mentioned in [46]. We used $128$ Monte Carlo (MC) samples from weight posterior for fair comparison with other stochastic methods in UQ benchmark [26], but we were able to get similar results with resuced number of MC samples.

**SVI-AvUC**  The model is trained with the same hyper-parameters and initializations with Empirical Bayes as described for SVI above, except that the model is trained with AvUC loss in addition to the ELBO loss. We used $\beta = 3$ in Equation 5 for relative weighting of AvUC loss with respect to ELBO loss.

**SVI-AvUTS**  We find the optimal temperature for pretrained SVI model by minimizing the *accuracy versus uncertainty calibration* (AvUC) loss on hold-out validation data. We adapted the code from [11] and replaced negative log-likelihood loss with our AvUC loss implementation. We used 50k images (randomly sampled from 1281.1k training images) for finding the optimal temperature to modify the logits of pretrained SVI. We used $128$ Monte Carlo samples from weight posterior for evaluation.

**AvUTS**  We applied AvUTS (AvU Temperature Scaling) method on pretrained vanilla ResNet-50 model with AvUC loss in order to compare with conventional temperature scaling [11] that optimizes negative log-likelihood loss. Results are provided in Appendix D.6. We used the pretrained model available in the torchvision package. We used entropy of softmax as uncertainty for AvUC loss

computation. We followed the same procedure as SVI-AvUTS described above, except that the method is applied to deterministic model.

We evaluate with 50k test images, along with 80 variants of dataset shift (each with 50k images) that includes 16 different types of datashift at 5 different intensities as described in Section A.

## C  Additional background

In this section, we follow the same notations described in Section 3.1 of the main paper.

### C.1  SVI in Bayesian deep neural networks

Bayesian deep neural networks provide a probabilistic interpretation of deep learning models by learning probability distributions over the neural network weights. In Bayesian setting, we would like to infer a distribution over weights w. A prior distribution is assumed over the weights $p(\mathrm{w})$ that captures our prior belief as to which parameters would have likely generated the outputs before observing any data. Given the evidence data $p(\mathrm{y}|\mathrm{x})$, prior distribution $p(\mathrm{w})$ and model likelihood $p(\mathrm{y} \mid \mathrm{x}, \mathrm{w})$, the goal is to infer the posterior distribution over the weights $p(\mathrm{w}|\mathrm{D})$:

$$p(\mathrm{w}|\mathrm{D}) = \frac{p(\mathrm{y} \mid \mathrm{x}, \mathrm{w})\, p(\mathrm{w})}{\int p(\mathrm{y} \mid \mathrm{x}, \mathrm{w})\, p(\mathrm{w})\, \mathrm{dw}} \tag{6}$$

Computing the posterior distribution $p(\mathrm{w}|\mathrm{D})$ is analytically intractable, stochastic variational inference (SVI) [2–4] is an approximate method that has been proposed to achieve tractable inference. SVI approximates a complex probability distribution $p(\mathrm{w}|\mathrm{D})$ with a simpler distribution $q_\theta(\mathrm{w})$, parameterized by variational parameters $\theta$ while minimizing the Kullback-Leibler (KL) divergence. Minimizing the KL divergence is equivalent to maximizing the log evidence lower bound (ELBO) [5], as given by Equation 7. Conventionally ELBO loss (negative ELBO) as given by Equation 8 is mizimized while training Bayesian deep neural networks with stochastic gradient descent optimization.

$$\mathbb{L} := \mathbb{E}_{q_\theta(\mathrm{w})}\left[\log p(\mathrm{y}|\mathrm{x}, \mathrm{w})\right] - KL[q_\theta(\mathrm{w})||p(\mathrm{w})] \tag{7}$$

$$\mathcal{L}_{\mathrm{ELBO}} := -\mathbb{E}_{q_\theta(\mathrm{w})}\left[\log p(\mathrm{y}|\mathrm{x}, \mathrm{w})\right] + KL[q_\theta(\mathrm{w})||p(\mathrm{w})] \tag{8}$$

In mean-field stochastic variation inference, weights are modeled with fully factorized Gaussian distribution parameterized by variational parameters $\mu$ and $\sigma$.

$$q_\theta(\mathrm{w}) = \mathcal{N}(\mathrm{w} \mid \mu, \sigma) \tag{9}$$

The variational distribution $q_\theta(\mathrm{w})$ and its parameters $\mu$ and $\sigma$ are learned while optimizing the cost function ELBO with the stochastic gradient steps.

### C.2  Uncertainty metrics

Predictive distribution is obtained through multiple stochastic forward passes on the network while sampling from the weight posteriors using Monte Carlo estimators. Equation 10 shows the predictive distribution of the output y given input x:

$$p(\mathrm{y}|\mathrm{x}, \mathrm{D}) \approx \frac{1}{T} \sum_{t=1}^{T} p(\mathrm{y}|\mathrm{x}, \mathrm{w_t}) , \quad \mathrm{w_t} \sim p(\mathrm{w} \mid \mathrm{D}) \tag{10}$$

**Predictive entropy**  The entropy [36] of the predictive distribution captures a combination of aleatoric and epistemic uncertainties [39] given by Equation 11 [35].

$$\mathbb{H}(\mathrm{y}|\mathrm{x}, \mathrm{D}) := -\sum_{k}\left(\frac{1}{T}\sum_{t=1}^{T} p\left(\mathrm{y} = k|\mathrm{x}, \mathrm{w_t}\right)\right) \log\left(\frac{1}{T}\sum_{t=1}^{T} p\left(\mathrm{y} = k|\mathrm{x}, \mathrm{w_t}\right)\right) \tag{11}$$

For deterministic models (Vanilla, Temp scaling), predictive entropy is computed with Equation 12.

$$\mathbb{H}(y|x, D) := -\sum_k \left( p\left(y = k|x, w\right)\right) \log \left( p\left(y = k|x, w\right)\right) \tag{12}$$

**Mutual information** The mutual information [36] between weight posterior and predictive distribution captures the epistemic uncertainty [35, 38] given by Equation 13.

$$MI(y, w|x, D) := \mathbb{H}(y|x, D) - \mathbb{E}_{p(w|D)}\left[\mathbb{H}(y|x, w)\right] \tag{13}$$

## C.3 Evaluation metrics

### C.3.1 Model calibration evaluation metrics

Expected calibration error (ECE) [40] measures the difference in expectation between model accuracy and its confidence as defined in Equation 14. ECE quantifies the model miscalibration with respect to confidence (probability of predicted class). The predictions of the neural network is partitioned into L bins of equal width, where $l^{th}$ bin is the interval $\left(\frac{l-1}{L}, \frac{l}{L}\right]$. ECE is computed using the equation below, where N is the total number of samples and $B_l$ is the set of indices of samples whose prediction confidence falls into the $l^{th}$ bin.

$$\text{ECE} = \sum_{l=1}^{L} \frac{|B_l|}{N} \left| \text{acc}\left(B_l\right) - \text{conf}\left(B_l\right)\right| \tag{14}$$

where the model accuracy and confidence per bin are defined as below.

$$\text{acc}\left(B_l\right) = \frac{1}{|B_l|} \sum_{i \in B_l} \mathbb{1}\left(\widehat{y}_i = y_i\right) \quad ; \quad \text{conf}\left(B_l\right) = \frac{1}{|B_l|} \sum_{i \in B_l} p_i \tag{15}$$

Expected uncertainty calibration error (UCE) [41] measures the difference in expectation between model error and its uncertainty as defined in Equation 16. UCE quantifies the model miscalibration with respect to predictive uncertainty representing entire predictive distribution of probabilities across the classes.

$$\text{UCE} = \sum_{l=1}^{L} \frac{|B_l|}{N} \left| \text{err}\left(B_l\right) - \text{uncert}\left(B_l\right)\right| \tag{16}$$

where the model error and uncertainty per bin are defined as below. $\tilde{u}_i \in [0, 1]$ represents normalized uncertainty.

$$\text{err}\left(B_l\right) = \frac{1}{|B_l|} \sum_{i \in B_l} \mathbb{1}\left(\widehat{y}_i \neq y_i\right) \quad ; \quad \text{uncert}\left(B_l\right) = \frac{1}{|B_l|} \sum_{i \in B_l} \tilde{u}_i \tag{17}$$

### C.3.2 Uncertainty evaluation metrics

Conditional probabilities *p(accurate | certain)* and *p(uncertain | inaccurate)* have been proposed in [39] as model performance evaluation metrics for comparing the quality of uncertainty estimates obtained from different probabilistic methods. *p(accurate | certain)* is given by Equation 18, measures the probability that the model is accurate on its output given that it is confident on the same. *p(uncertain | inaccurate)* is given by Equation 19, measures the probability that the model is uncertain about its output given that it has made inaccurate prediction.

$$p(\text{accurate}|\text{certain}) = \frac{n_{AC}}{n_{AC} + n_{IC}} \tag{18}$$

$$p(\text{uncertain}|\text{inaccurate}) = \frac{n_{IU}}{n_{IC} + n_{IU}} \tag{19}$$

# D Additional Results

## D.1 Monitoring metrics and loss functions while training with SVI-AvUC

Figure F3: SVI-AvUC ResNet-20/CIFAR: Training. Monitoring accuracy, AvU metric, ELBO loss, AvUC loss and total loss at each training epoch.

Figure F4: SVI-AvUC ResNet-20/CIFAR: Validation accuracy and AvU score. Monitoring accuracy and AvU metric on test data at after each training epoch.

Figure F3 shows ELBO loss, AvUC (*acuuracy vs uncertainty calibration*) loss and total loss (combination of ELBO and AvUC losses) along with accuracy and AvU metrics at each training epoch. ELBO loss consist of two components including negative expected log-likelihood and Kullback-Leibler divergence as given by Equation 8. We can observe that the ELBO loss decreases as accuracy is increasing indicating the inverse correlation between them. We can also see that ELBO loss is decreasing even if the AvU score is not increasing. AvU provides relationship between accuracy and uncertainty that hints model calibration as described in Section 3. Figure F3(b) and (d) show that the proposed differentiable AvUC loss and actual AvU metric is inversely correlated, guiding the gradient optimization of total loss with respect to improving both accuracy and uncertainty calibration. Figure F4 shows accuracy and AvU score on test data obtained from 1 Monte Carlo sample at the end of each training epoch (for monitoring). The model accuracy and AvU score during evaluation phase will be higher as we use larger number of Monte Carlo samples to marginalize over the weight posterior.

## D.2 Additional results for model calibration evaluation

In addition to model calibration evaluation with *expected calibration error* (ECE) ↓ and *expected uncertainty calibration error* (UCE) ↓ metrics in Figure 1 of Section 4, we also compare negative log-likelood (NLL) ↓ and Brier score metrics ↓ obtained from different methods on ImageNet (ResNet-50) and CIFAR10 (ResNet-20) across 80 combinations of datashift including 16 different types of shift at 5 different levels of shift intensities. The results are shown in Figure F5 for ImageNet and in Figure F6 for CIFAR10.

Figure F5: ResNet-50/ImageNet: Model calibration comparison using ECE↓, UCE↓, NLL↓ and Brier score↓ on ImageNet under in-distribution (test) and dataset shift at different levels of shift intensities (1-5). A well-calibrated model should consistently provide lower ECE, UCE, NLL and Brier score even at increased levels of datashift, as accuracy may degrade with increased datashift. At each shift intensity level, the boxplot summarizes the results across 16 different datashift types showing the min, max, mean and quartiles.

Figure F6: ResNet-20/CIFAR10: Model calibration comparison using ECE↓, UCE↓, NLL↓ and Brier score↓ on CIFAR10 under in-distribution (test) and dataset shift at different levels of shift intensities (1-5). A well-calibrated model should consistently provide lower ECE, UCE, NLL and Brier score even at increased levels of datashift, as accuracy may degrade with increased datashift. At each shift intensity level, the boxplot summarizes the results across 16 different datashift types showing the min, max, mean and quartiles.

The Spearman rank-order correlation coefficient ($\rho$) [55] is a nonparametric measure of rank correlation, which assesses the monotonic relationships between two variables. Spearman's $\rho \in [-1, 1]$, with -1 or +1 implies exact monotonic relationship (negative and positive correlations respectively) and 0 implies no correlation between two variables. We assess the effect of increasing data shift intensities on the model calibration errors with Spearman rank-order correlation coefficient as shown in Table T1. A perfectly calibrated and robust model will have Spearman's $\rho$ equal to 0 indicating the model calibration errors are not correlated to data shift. The results in Table T1 shows that ECE and UCE increases with data shift for all the methods, with comparatively lower $\rho$ values for SVI-AvUC indicating the proposed method is robust to data shift.

Table T1: Spearman rank-order correlation coefficient assessing the monotonic relationship between model calibration errors (ECE and UCE) and the data shift intensity for the results in the Figures F5 and F6. Spearman's $\rho$ indicates SVI-AvUC is robust as model calibration errors are less correlated to data shift compared to other methods. $\rho$ value near to 0 is better.

| Dataset/Model | Spearman's $\rho$ rank-order correlation co-eff wrt dataset shift intensity | Method | | | | | | | | | |
|---|---|---|---|---|---|---|---|---|---|---|---|
| | | Vanilla | Temp scaling | Ensemble | Dropout | LL Dropout | SVI | LL-SVI | SVI-TS | SVI-AvUTS | SVI-AvUC |
| ImageNet/ | $\rho_{ECE}$ | 1.0 | 1.0 | 0.6 | 1.0 | 1.0 | 1.0 | 1.0 | 1.0 | 0.94 | **0.31** |
| ResNet-50 | $\rho_{UCE}$ | 1.0 | 1.0 | 1.0 | 1.0 | 1.0 | 1.0 | 1.0 | 1.0 | 1.0 | **0.94** |
| CIFAR10/ | $\rho_{ECE}$ | 1.0 | 1.0 | 1.0 | 0.94 | 1.0 | 1.0 | 1.0 | 0.94 | 0.82 | **0.71** |
| ResNet-20 | $\rho_{UCE}$ | 1.0 | 1.0 | 1.0 | 1.0 | 1.0 | 1.0 | 1.0 | 0.77 | 0.82 | **0.71** |

Table T2: ImageNet: calibration under distributional shift. The lower quartile(25th percentile), median (50th percentile), mean and upper quartile (75th percentile) of ECE ↓, UCE ↓, NLL ↓ and Brier score ↓ computed across 16 different types of datashift at intensity 5 are presented below.

| Metric | | Methods | | | | | | | | |
|---|---|---|---|---|---|---|---|---|---|---|
| | | Vanilla | Temp scaling | Ensemble | Dropout | LL Dropout | SVI | LL SVI | SVI-AvUTS | SVI-AvUC |
| **ECE ↓** | lower quartile | 0.1244 | 0.0959 | 0.0503 | 0.0783 | 0.0925 | 0.0722 | 0.1212 | 0.0420 | 0.0319 |
| | median | 0.1737 | 0.1392 | 0.0900 | 0.1339 | 0.1450 | 0.1144 | 0.1684 | 0.0807 | 0.0447 |
| | mean | 0.1942 | 0.1600 | 0.0880 | 0.1530 | 0.1612 | 0.1188 | 0.1868 | 0.0800 | 0.0542 |
| | upper quartile | 0.2744 | 0.2364 | 0.1264 | 0.2186 | 0.2364 | 0.1723 | 0.2676 | 0.1275 | 0.0696 |
| **UCE ↓** | lower quartile | 0.3068 | 0.2701 | 0.2179 | 0.2552 | 0.2727 | 0.2125 | 0.3356 | 0.1725 | 0.1310 |
| | median | 0.3664 | 0.3251 | 0.2848 | 0.3506 | 0.3427 | 0.2872 | 0.3817 | 0.2323 | 0.1853 |
| | mean | 0.3826 | 0.3428 | 0.2813 | 0.3651 | 0.3593 | 0.2865 | 0.4007 | 0.2263 | 0.1774 |
| | upper quartile | 0.4752 | 0.4335 | 0.3506 | 0.4511 | 0.4572 | 0.3587 | 0.4917 | 0.2901 | 0.2113 |
| **NLL ↓** | lower quartile | 4.635 | 4.530 | 4.035 | 4.699 | 4.563 | 4.322 | 5.417 | 4.278 | 4.164 |
| | median | 5.115 | 4.993 | 4.624 | 5.093 | 5.034 | 4.853 | 6.076 | 4.912 | 4.823 |
| | mean | 5.234 | 5.091 | 4.604 | 5.553 | 5.201 | 4.865 | 6.422 | 4.860 | 4.707 |
| | upper quartile | 6.292 | 6.165 | 5.893 | 6.522 | 6.342 | 6.034 | 7.755 | 5.941 | 5.778 |
| **Brier score ↓** | lower quartile | 0.941 | 0.926 | 0.877 | 0.933 | 0.923 | 0.906 | 0.963 | 0.893 | 0.883 |
| | median | 0.987 | 0.970 | 0.922 | 0.967 | 0.969 | 0.943 | 0.998 | 0.948 | 0.935 |
| | mean | 0.964 | 0.945 | 0.888 | 0.961 | 0.947 | 0.922 | 0.979 | 0.914 | 0.900 |
| | upper quartile | 1.052 | 1.027 | 0.989 | 1.025 | 1.025 | 1.013 | 1.072 | 0.996 | 0.985 |

Table T3: CIFAR10: calibration under distributional shift. The lower quartile(25th percentile), median (50th percentile), mean and upper quartile (75th percentile) of ECE ↓, UCE ↓, NLL ↓ and Brier score ↓ computed across 16 different types of datashift at intensity 5 are presented below.

| Metric | | Methods | | | | | | | | | |
|---|---|---|---|---|---|---|---|---|---|---|---|
| | | Vanilla | Temp scaling | Ensemble | Dropout | LL Dropout | SVI | LL SVI | Radial BNN | SVI-AvUTS | SVI-AvUC |
| **ECE ↓** | lower quartile | 0.2121 | 0.0997 | 0.0549 | 0.0794 | 0.2022 | 0.0925 | 0.2027 | 0.0797 | 0.0466 | 0.0398 |
| | median | 0.3022 | 0.1834 | 0.1045 | 0.1889 | 0.3643 | 0.2146 | 0.3077 | 0.1950 | 0.1516 | 0.1107 |
| | mean | 0.3151 | 0.1993 | 0.1611 | 0.2405 | 0.3518 | 0.2389 | 0.3267 | 0.2150 | 0.1585 | 0.1374 |
| | upper quartile | 0.4148 | 0.2915 | 0.2551 | 0.3518 | 0.4854 | 0.3636 | 0.4246 | 0.3410 | 0.2345 | 0.2303 |
| **UCE ↓** | lower quartile | 0.1813 | 0.0419 | 0.0417 | 0.0328 | 0.1728 | 0.0594 | 0.1875 | 0.0473 | 0.0575 | 0.0495 |
| | median | 0.2773 | 0.1147 | 0.0653 | 0.1382 | 0.3336 | 0.1723 | 0.2747 | 0.1449 | 0.11486 | 0.0740 |
| | mean | 0.2853 | 0.1429 | 0.1333 | 0.1974 | 0.3204 | 0.2008 | 0.2983 | 0.1741 | 0.1272 | 0.1038 |
| | upper quartile | 0.3871 | 0.2232 | 0.2103 | 0.2903 | 0.4486 | 0.3034 | 0.3902 | 0.2941 | 0.1827 | 0.1512 |
| **NLL ↓** | lower quartile | 1.634 | 1.166 | 0.955 | 0.971 | 1.419 | 1.052 | 1.629 | 1.179 | 0.984 | 1.035 |
| | median | 2.666 | 1.957 | 1.753 | 1.952 | 2.767 | 2.001 | 2.752 | 2.038 | 1.747 | 1.742 |
| | mean | 2.653 | 1.846 | 1.779 | 2.036 | 2.682 | 2.017 | 2.764 | 1.995 | 1.728 | 1.633 |
| | upper quartile | 3.617 | 2.467 | 2.587 | 2.652 | 3.780 | 2.952 | 3.762 | 2.706 | 2.507 | 2.158 |
| **Brier score ↓** | lower quartile | 0.546 | 0.496 | 0.407 | 0.421 | 0.526 | 0.449 | 0.529 | 0.488 | 0.434 | 0.454 |
| | median | 0.871 | 0.765 | 0.651 | 0.727 | 0.848 | 0.702 | 0.850 | 0.738 | 0.675 | 0.692 |
| | mean | 0.785 | 0.697 | 0.639 | 0.728 | 0.820 | 0.702 | 0.803 | 0.719 | 0.657 | 0.646 |
| | upper quartile | 0.995 | 0.876 | 0.844 | 0.943 | 1.111 | 0.957 | 1.017 | 0.960 | 0.876 | 0.837 |

## D.3 Additional results for confidence and uncertainty evaluation under distributional shift

(a) Defocus blur

(b) Glass blur

Figure F7: ImageNet: Model confidence and uncertainty evaluation under distributional shift (defocus blur and glass blur of intensity 3). Column 1: accuracy as a function of confidence. We expect a reliable model to be more accurate at higher confidence values; Column 2: number of examples above given confidence value. We expect a reliable model to have lesser number of examples with higher confidence as accuracy is significantly degraded under distributional shift; Column 3: probability of model being uncertain when making inaccurate predictions. We expect a reliable model to be more uncertain when it is inaccurate. Normalized uncertainty thresholds $t \in [0, 1]$ are shown in plots as the uncertainty range varies for different methods. All the plots show SVI-AvUC outperforms other methods.

(a) Speckle noise

(b) Shot noise

Figure F8: CIFAR: Model confidence and uncertainty evaluation under distributional shift (speckle noise and shot noise of intensity 3). Column 1: accuracy as a function of confidence; Column 2: probability of model being accurate on its predictions when it is certain; Column 3: probability of model being uncertain when making inaccurate predictions. Normalized uncertainty thresholds $t \in [0, 1]$ are shown in plots as the uncertainty range varies for different methods. All the plots show SVI-AvUC outperforms other methods.

## D.4 Comparing AUC of accuracy vs uncertainty (AvU) measures

Figure F9: ImageNet: AvU AUC↑ on in-distribution (test) and under dataset shift at different levels of shift intensities (1-5). We expect a well-calibrated model to consistently provide higher AvU AUC score even at increased levels of datashift. At each shift intensity level, the boxplot summarizes the results across 16 different datashift types showing the min, max and quartiles. SVI-AvUC and SVI-AvUTS yields higher area under the curve of AvU (AvU AUC) computed across various uncertainty thresholds at increased data shift intensity.

Table T4: Spearman rank-order correlation coefficient assessing the relationship between AvU-AUC and data shift intensity. Spearman's $\rho$ indicates that AUC of AvU degrades with increased data shift for all the methods with comparatively SVI-AvUC being robust ($\rho$ value near to 0 is better).

| Spearman's $\rho$ rank-order correlation coeff wrt data shift intensity | Method | | | | | | | | | |
|---|---|---|---|---|---|---|---|---|---|---|
| | Vanilla | Temp scaling | Ensemble | Dropout | LL Droput | SVI | LL SVI | SVI-TS | SVI-AvUTS | SVI-AvUC |
| $\rho_{AvUAUC}$ | -1.0 | -0.94 | -0.82 | -0.94 | -1.0 | -0.82 | -1.0 | -0.82 | -0.6 | **-0.25** |

## D.5 Addition results for distributional shift detection

Figure F10 shows the density histogram plots of predictive uncertainty estimates obtained from different methods on SVHN dataset (out-distribution) and CIFAR10 test set (in-distribution) with ResNet-20 model that trained with CIFAR-10. These plots correspond to the out-of-distribution detection results presented in Table 2 of Section 4.

Figure F10: Out-of-distribution SVHN: Density histograms of predictive entropy on SVHN as OOD and CIFAR10 as in-distribution (ResNet-20 trained with CIFAR10). SVI-AvUC shows best separation of entropy densities between in-distribution and OOD as quantified by Wasserstein distance in Table T5.

Table T5: Wasserstein distance between the distribution of predictive uncertainties on CIFAR10 test data (in-distribution) and SVHN data (out-of-distribution).

| Method | Wasserstein distance |
|---|---|
| Vanilla | 0.6703 |
| Temp scaling | 0.9350 |
| Ensemble | 0.9043 |
| Dropout | 0.6767 |
| LL Dropout | 0.4905 |
| Radial BNN | 0.3933 |
| SVI | 0.7480 |
| LL SVI | 0.6367 |
| SVI-TS | 0.7874 |
| SVI-AvUTS | 0.8469 |
| SVI-AvUC | **1.2021** |

Table T6: Wasserstein distance between the distribution of predictive uncertainties on ImageNet test data(in-distribution) and data shifted with defocus blur at intensity 5.

| Method | Wasserstein distance |
|---|---|
| Vanilla | 3.0173 |
| Temp scaling | 3.1866 |
| Ensemble | 3.2473 |
| Dropout | 3.2605 |
| LL Dropout | 3.3676 |
| SVI | 3.6339 |
| LL SVI | 2.9897 |
| SVI-TS | 3.6851 |
| SVI-AvUTS | 3.9466 |
| SVI-AvUC | **4.2043** |

Figure F11 shows the density histogram plots of predictive uncertainty estimates obtained from different methods on ImageNet test set (in-dist) and defocus blur of intensity 5 (data shift) with ResNet-50 model that was trained with clean ImageNet.

Figure F11: Data shift on ImageNet (defocus blur): Density histograms of predictive entropy on ImageNet in-distribution test set and data shifted with defocus blur (ResNet-50 trained with clean ImageNet). SVI-AvUC shows best separation of entropy densities between in-distribution and data-shift. SVI-AvUC shows best separation of predictive uncertainty densities between in-distribution and shifted data as quantified by Wasserstein distance in Table T6.

Table T7 provides comprehensive distributional shift detection performance evaluation of different methods across 16 different types of datashift at intensity 5 on ImageNet as described in Section A. We observe SVI-AvUC performing best in detecting most of the shift types, and Ensemble perform best on few of the shift types.

Table T7: ImageNet: Distributional shift detection using predictive entropy. We compare distributional shift detection performance on 16 different types of dataset shift(each type contains 50k shifted test images). All values are in percentages and best results are indicated in bold.

| Dataset shift type | Detection evaluation metric ↑ | Methods | | | | | | | | |
|---|---|---|---|---|---|---|---|---|---|---|
| | | Vanilla | Temp scaling | Ensemble | Dropout | LL Dropout | SVI | LL SVI | SVI-AvUTS | SVI-AvUC |
| **Gaussian blur** | AUROC | 93.36 | 93.71 | 95.49 | 96.38 | 96.04 | 96.40 | 93.58 | 96.89 | **97.60** |
| | Det. accuracy | 86.08 | 86.47 | 88.82 | 89.98 | 89.68 | 90.03 | 86.93 | 90.93 | **92.07** |
| | AUPR-in | 92.82 | 93.21 | 95.31 | 96.16 | 95.63 | 95.97 | 92.06 | 96.58 | **97.39** |
| | AUPR-out | 93.71 | 94.01 | 95.64 | 96.67 | 96.40 | 96.83 | 94.02 | 97.19 | **97.85** |
| **Brightness** | AUROC | 70.58 | 71.02 | 71.97 | 73.73 | 71.17 | 72.77 | 69.24 | **75.08** | 74.61 |
| | Det. accuracy | 65.03 | 65.45 | 66.15 | 67.44 | 65.36 | 66.61 | 64.16 | **68.44** | 67.58 |
| | AUPR-in | 68.28 | 68.62 | 70.57 | 72.42 | 68.96 | 70.93 | 65.41 | **73.12** | 73.54 |
| | AUPR-out | 70.80 | 71.26 | 71.62 | 73.75 | 71.48 | 73.34 | 69.60 | **75.93** | 75.56 |
| **Contrast** | AUROC | 98.82 | 98.96 | 99.40 | 99.41 | 99.32 | 98.92 | 98.73 | 99.45 | **99.48** |
| | Det. accuracy | 94.70 | 95.06 | 96.27 | 96.22 | 96.06 | 94.87 | 94.59 | 96.52 | **96.69** |
| | AUPR-in | 98.75 | 98.91 | 99.39 | 99.41 | 99.28 | 98.85 | 98.64 | 99.44 | **99.46** |
| | AUPR-out | 98.91 | 99.04 | 99.42 | 99.43 | 99.37 | 99.02 | 98.85 | 99.48 | **99.52** |
| **Defocus blur** | AUROC | 94.04 | 94.37 | 95.74 | 96.26 | 95.97 | 95.88 | 93.69 | 96.68 | **97.18** |
| | Det. accuracy | 86.79 | 87.13 | 89.06 | 89.79 | 89.52 | 89.35 | 86.98 | 90.51 | **91.40** |
| | AUPR-in | 93.34 | 93.70 | 95.40 | 96.03 | 95.44 | 95.37 | 92.01 | 96.29 | **96.91** |
| | AUPR-out | 94.66 | 94.94 | 96.11 | 96.58 | 96.43 | 96.39 | 94.28 | 97.05 | **97.50** |
| **Elastic transform** | AUROC | 88.15 | 88.81 | **91.03** | 87.73 | 89.20 | 89.63 | 86.73 | 90.84 | 90.82 |
| | Det. accuracy | 80.43 | 81.16 | **83.59** | 80.19 | 81.69 | 82.12 | 79.44 | 83.28 | 83.06 |
| | AUPR-in | 88.45 | 89.10 | **91.43** | 88.57 | 89.56 | 89.99 | 86.08 | 91.08 | 91.29 |
| | AUPR-out | 87.18 | 87.84 | 90.06 | 85.97 | 88.01 | 88.58 | 86.19 | 90.07 | **90.08** |
| **Fog** | AUROC | 89.15 | 89.74 | 91.45 | 91.83 | 90.03 | 90.20 | 87.40 | **93.01** | 91.46 |
| | Det. accuracy | 81.12 | 81.79 | 83.78 | 84.00 | 82.03 | 82.48 | 79.75 | **85.47** | 83.44 |
| | AUPR-in | 88.75 | 89.30 | 91.39 | 92.04 | 89.85 | 89.90 | 85.78 | **92.84** | 91.13 |
| | AUPR-out | 89.22 | 89.83 | 91.34 | 91.61 | 89.99 | 90.26 | 87.67 | **93.14** | 91.90 |
| **Frost** | AUROC | 88.67 | 89.19 | 90.90 | 90.53 | 88.56 | 90.60 | 87.69 | 91.74 | **92.19** |
| | Det. accuracy | 80.87 | 81.40 | 83.23 | 82.64 | 80.65 | 82.84 | 80.07 | 83.99 | **84.31** |
| | AUPR-in | 87.95 | 88.46 | 90.56 | 90.44 | 87.80 | 89.91 | 86.09 | 91.03 | **91.63** |
| | AUPR-out | 89.03 | 89.55 | 91.06 | 90.56 | 88.98 | 91.20 | 88.10 | 92.41 | **92.88** |
| **Glass blur** | AUROC | 94.96 | 95.29 | 96.48 | 96.06 | 96.02 | 96.90 | 95.14 | 97.37 | **97.85** |
| | Det. accuracy | 87.86 | 88.31 | 90.15 | 89.41 | 89.40 | 90.71 | 88.68 | 91.58 | **92.51** |
| | AUPR-in | 94.71 | 95.06 | 96.32 | 95.94 | 95.76 | 96.68 | 94.28 | 97.20 | **97.70** |
| | AUPR-out | 95.24 | 95.54 | 96.66 | 96.24 | 96.26 | 97.17 | 95.52 | 97.57 | **98.05** |
| **Gaussian noise** | AUROC | 92.36 | 92.84 | **97.78** | 91.27 | 93.87 | 95.83 | 91.00 | 96.37 | 97.46 |
| | Det. accuracy | 85.25 | 85.92 | **92.92** | 85.84 | 87.31 | 89.29 | 84.60 | 90.10 | 91.73 |
| | AUPR-in | 92.66 | 93.16 | **97.97** | 93.39 | 94.60 | 95.91 | 91.75 | 96.44 | 97.46 |
| | AUPR-out | 91.20 | 91.62 | 97.42 | 86.10 | 92.70 | 95.76 | 89.03 | 96.28 | **97.52** |
| **Impulse noise** | AUROC | 92.15 | 92.63 | **97.64** | 92.10 | 93.77 | 95.39 | 91.68 | 96.01 | 97.14 |
| | Det. accuracy | 85.03 | 85.69 | **92.76** | 86.81 | 87.10 | 88.73 | 85.04 | 89.56 | 91.17 |
| | AUPR-in | 92.59 | 93.09 | **97.91** | 94.01 | 94.44 | 95.51 | 92.27 | 96.10 | 97.20 |
| | AUPR-out | 90.75 | 91.17 | 97.15 | 86.95 | 92.67 | 95.25 | 90.05 | 95.87 | **97.17** |
| **Pixelate** | AUROC | 81.52 | 81.88 | 87.80 | 88.03 | 87.01 | 87.98 | 79.85 | 87.19 | **90.04** |
| | Det. accuracy | 74.37 | 74.71 | 80.23 | 80.64 | 79.50 | 80.24 | 73.17 | 79.27 | **81.98** |
| | AUPR-in | 80.02 | 80.39 | 87.16 | 87.94 | 86.03 | 87.07 | 76.98 | 86.10 | **89.48** |
| | AUPR-out | 81.34 | 81.66 | 87.56 | 86.91 | 86.93 | 88.27 | 79.56 | 87.52 | **90.48** |
| **Saturate** | AUROC | 74.37 | 74.83 | 76.70 | 75.70 | 74.19 | 77.21 | 73.26 | 78.05 | **78.71** |
| | Det. accuracy | 68.32 | 68.79 | 70.37 | 69.60 | 68.22 | 70.65 | 67.52 | 71.41 | **71.57** |
| | AUPR-in | 71.66 | 72.04 | 74.53 | 74.21 | 71.54 | 74.95 | 69.38 | 75.84 | **77.31** |
| | AUPR-out | 73.84 | 74.29 | 75.90 | 73.75 | 73.24 | 77.07 | 72.94 | 77.76 | **78.82** |
| **Shot noise** | AUROC | 90.38 | 90.92 | **97.15** | 90.31 | 93.25 | 95.15 | 90.29 | 95.57 | 96.72 |
| | Det. accuracy | 83.11 | 83.79 | **91.98** | 84.88 | 86.74 | 88.49 | 84.02 | 89.02 | 90.47 |
| | AUPR-in | 90.72 | 91.29 | **97.41** | 92.55 | 94.07 | 95.27 | 91.03 | 95.68 | 96.77 |
| | AUPR-out | 88.86 | 89.32 | 96.53 | 84.79 | 91.68 | 95.00 | 87.45 | 95.38 | **96.75** |
| **Spatter** | AUROC | 84.23 | 84.92 | **88.01** | 84.87 | 84.60 | 86.01 | 83.41 | 87.00 | 86.34 |
| | Det. accuracy | 76.74 | 77.49 | **80.78** | 77.95 | 77.66 | 78.75 | 76.15 | 79.53 | 78.73 |
| | AUPR-in | 84.06 | 84.69 | **88.05** | 85.73 | 84.66 | 85.81 | 81.38 | 86.59 | 86.61 |
| | AUPR-out | 82.90 | 83.62 | **86.66** | 81.50 | 82.38 | 84.90 | 83.21 | 86.36 | 85.12 |
| **Speckle noise** | AUROC | 87.32 | 87.83 | **93.17** | 88.54 | 88.54 | 90.28 | 87.13 | 90.58 | 91.84 |
| | Det. accuracy | 80.05 | 80.64 | **86.40** | 82.00 | 82.00 | 82.87 | 80.39 | 83.10 | 84.02 |
| | AUPR-in | 87.57 | 88.09 | **93.32** | 89.88 | 89.88 | 90.17 | 86.89 | 90.41 | 91.88 |
| | AUPR-out | 85.38 | 85.84 | **92.25** | 84.48 | 84.48 | 89.78 | 84.78 | 90.12 | 91.70 |
| **Zoom blur** | AUROC | 89.92 | 90.48 | 92.12 | 90.47 | 90.77 | 90.65 | 88.65 | 91.56 | **93.87** |
| | Det. accuracy | 82.29 | 82.92 | 84.79 | 82.85 | 83.32 | 83.11 | 81.36 | 84.14 | **86.62** |
| | AUPR-in | 88.84 | 89.40 | 91.49 | 90.18 | 89.86 | 89.82 | 86.31 | 91.01 | **93.41** |
| | AUPR-out | 90.39 | 90.93 | 92.36 | 90.27 | 91.07 | 91.04 | 89.01 | 91.68 | **94.40** |

### D.6 AvUTS applied to Vanilla DNN (Comparison with Temp scaling using NLL)

We evaluate AvUTS (AvU Temperature Scaling) by performing post-hoc calibration on vanilla DNN with *accuracy versus uncertainty calibration* (AvUC) loss and compare with conventional temperature scaling [11] that optimizes negative log-likelihood loss. We use entropy of softmax as uncertainty for AvUC loss computation.

Figure F12: AvUTS on Vanilla ResNet-50: Model calibration comparison of AvUTS with conventional Temp Scaling and Vanilla baselines using ECE↓ and UCE↓ on ImageNet under in-distribution (test) and dataset shift at different levels of shift intensities (1-5). A well-calibrated model should provide lower calibration errors even at increased levels of datashift, though accuracy may degrade with data shift. At each shift intensity level, the boxplot summarizes the results across 16 different datashift types showing the min, max and quartiles. We can see that AvUTS provides significantly lower model calibration errors (ECE and UCE) than Vanilla and Temp scaling methods at increased distributional shift intensity, while providing comparable accuracy.

## E  Ablation study for $\beta$ weight factor in SVI-AvUC

Figure F13: Model calibration errors (ECE, UCE) and accuracy at different values of $\beta$ in Equation 5

We evaluate SVI-AvUC method on ResNet-20 model with different values of $\beta$ in Equation 5. Figure F13 shows the effect of different values of $\beta$ on the model calibration errors (ECE and UCE) and model test accuracy on test data shifted with Gaussian blur at intensity 3. We observe that the accuracy curve remains almost flat with different $\beta$ values, ECE decreases initially and increases above $\beta=3$, UCE decreases initially with $\beta$ and then remains almost flat.

# F  Optimizing Area under the curve of AvU

We optimized area under the curve of AvU across various uncertainty thresholds towards a threshold free mechanism. This method is compute intensive during training as we need to compute AvU at different thresholds $u_{th} = u_{\min} + (t\,(u_{\max} - u_{\min}))$ with $t \in [0, 1]$. We applied this method to both training the model and post-hoc calibration on SVI (SVI-AUAvUC and SVI-AUAvUTS), results are shown in Figure F14. The results are similar to SVI-AvUC and SVI-AvUTS as presented in Figure F6.

Figure F14: AUC of AvU optimized ResNet-20/CIFAR10: Model calibration comparison using ECE↓ on CIFAR10 under in-distribution (test) and dataset shift at different levels of shift intensities (1-5). A well-calibrated model should consistently provide lower calibration error even at increased levels of datashift, though accuracy may degrade with increased datashift. At each shift intensity level, the boxplot summarizes the results across 16 different datashift types showing the min, max and quartiles.

## Footnotes

[1] `https://github.com/pytorch/vision/blob/master/torchvision/models/resnet.py`