[Reviews · NeurIPS 2020]

Review 1

Summary and Contributions: This manuscript proposes a formulation that leverages the relationship between accuracy and uncertainty to improve model calibration, as well as optimization strategies and a post-processing method to apply to both end-to-end training and pre-trained models.

Strengths: Overall, the method this manuscript propose is convincing. The formulation is intuitive and model agnostic. Proposed training algorithm is a more computationally efficient compared with model ensemble and provides better uncertainty calibration under various distributional shifts compared with approximate Bayesian inference. Empirical results are promising and shows the AvUC loss can help to balance between the accuracy and different uncertainty calibration metrics.

Weaknesses: 1. The ensemble method looks like a much stronger baseline than others. It would be better to demonstrate the superiority over some baseline algorithms both empirically and theoretically. 2. I would be more interested to see how the algorithm will perform in regression problems. Similar to the work of e.g. conformal prediction https://arxiv.org/pdf/1905.03222.pdf can we obtain high quality uncertainty bounds from this algorithm?

Correctness: Both the claim and empirical methodology are correct.

Clarity: The paper is well written.

Relation to Prior Work: The authors clearly discussed the drawbacks of previous works and proposed a different approach to solve this problem. However, it would be better to add a discussion section on the superiority/optimality of AvUC loss over other methods before the empirical evaluations.

Reproducibility: Yes

Additional Feedback: The authors have well addressed the concerns. For the ensemble baseline, most of the model ensemble methods (e.g., simple bagging approach) will result in over-confident predictions [1, 2], which might be a good direction to demonstrate the superiorities. [1] Lakshminarayanan et al. Simple and Scalable Predictive Uncertainty Estimation using Deep Ensembles, NIPS 2017 [2] Li et al. Improving Confidence Estimates for Unfamiliar Examples.


Review 2

Summary and Contributions: This paper considers the problem of reliable and accurate quantification of uncertainty estimates from deep neural networks. The authors propose accuracy versus uncertainty calibration (AvUC) loss function and used it to improve the performance of stochastic variational inference (SVI) based BNN.

Strengths: The approach is simple to use with existing frameworks. Empirical results are comprehensive and show better performance compared to existing methods.

Weaknesses: No theoretical justification and analysis is provided for the proposed approach.

Correctness: Yes

Clarity: Yes

Relation to Prior Work: Yes

Reproducibility: Yes

Additional Feedback: Post Rebuttal: The authors have satisfactorily addressed all my concerns. Specifically, my major concern on the absence of theoretical justification behind this approach will be addressed by authors incorporating R4's suggestion on discussing how the approach serves as loss-calibrated inference method. This would certainly make this paper strong. ++++++++++++++++++++++++++++++++++++++++++++++++++++++++++++ In this paper, the authors propose a modified loss function for improving the performance of uncertainty-aware DNNs. They show applicability of their loss with mean-field stochastic variational inference (SVI) based BNN. The formulation is interesting, however, there are some issues needed to be resolved before any claim can be made regarding usefulness of this approach. My concerns are explained below: The proposed loss function is a straightforward extension of the patch accuracy versus patch uncertainty (PAvPU) loss from Mukhoti and Gal [35]. Later, it is simply used in the SVI framework. I would encourage authors to provide more insight into this loss functions. For example, what is the theoretical justification of eq. (5)? Without any theoretical analysis of this modified loss function, the work seems to be incremental. I would also encourage authors to look into some recent papers [1,2] questioning the reliability of the calibration metrics and use suggested evaluation metrics. Being said that, I believe this is an interesting formulation with some strong empirical results. This approach can be potentially useful and impactful, however, requires more work. Thus, I will encourage authors to make it a little more mature by addressing some concerns as mentioned above. [1] Kumar, Ananya, Percy Liang, and Tengyu Ma. "Verified Uncertainty Calibration." arXiv preprint arXiv:1909.10155 (2019). [2] Zhang, Jize, Bhavya Kailkhura, and T. Han. "Mix-n-Match: Ensemble and Compositional Methods for Uncertainty Calibration in Deep Learning." arXiv preprint arXiv:2003.07329 (2020).


Review 3

Summary and Contributions: The paper proposes a novel loss measure AvUC to improve calibration of classifiers. The authors illustrate its benefits in combination with the existing mean-field stochastic variational inference (SVI) probabilistic classification method. In particular, two methods are proposed: SVI-AvUC uses the novel loss measure as an additional loss term in SVI optimization; and SVI-AvUTS transforms SVI predictions with a modified temperature scaling method where the temperature is chosen to minimize AvUC instead of cross entropy. The results show improvements over SVI as well as over vanilla DNN and its standard variations (temperature scaling, dropout, last layer dropout, deep-ensembles).

Strengths: * A strong related work section. * Good results for the proposed methods in empirical experiments against several existing methods. * Well-written text and detailed supplementary.

Weaknesses: * The proposed methods have not been compared against simple temperature scaling on top of the SVI method (could be denoted SVI-TS). SVI-TS would be a natural competitor for SVI-AvUTS, because SVI-AvUTS is essentially obtained from SVI-TS after replacing cross-entropy by the proposed AvUC loss. * Comparisons have been made only on top of SVI, and the usefulness of the loss beyond this particular scenario has not been studied. * Not sufficiently explained why the proposed loss should be better than existing losses. * Not investigated whether AvUC is a proper loss.

Correctness: The claims about the advantages of AvUC might be too bold since the comparisons against SVI-TS (SVI+standard temperature scaling) have not been performed.

Clarity: The paper is mostly well-written. Minor issues will be highlighted in below additional feedback.

Relation to Prior Work: Yes, it is clear how this work differs from previous contributions.

Reproducibility: Yes

Additional Feedback: The choice of only evaluating the proposed methods on top of SVI is problematic, in the sense that the usefulness of the loss beyond this particular scenario remains unstudied (except for supplementary figure F12). It seems like the authors have worked towards getting as good results as possible, but have not made enough effort towards finding out which factors contribute towards these results and whether simpler alternatives with similar performance exist. In particular, the proposed methods have not been compared against simple temperature scaling on top of the SVI method (could be denoted SVI-TS). SVI-TS would be a natural competitor for SVI-AvUTS, because SVI-AvUTS is essentially obtained from SVI-TS after replacing cross-entropy by the proposed AvUC loss. The supplementary does compare AvUTS and TS on top of a vanilla ResNet50 showing some improvements of AvUTS over TS on high data shift intensities. However, it is not clear to which extent such improvements would carry over to the SVI scenario where the performance is already better than vanilla ResNet50. While the overall intuition behind AvUC has been explained, the technical details have not been justified. What makes this loss better than existing proper losses (cross entropy, Brier score) or trainable calibration losses (MMCE [11]). Given the potentially similar usage, comparisons of AvUC against MMCE would also have been desirable. The claim 'Existing calibration methods do not explicitly account for quality of predictive uncertainty estimates while training the model' at line 89 seems wrong to me due to the MMCE methodology proposed in [11]. Is AvUC a proper loss? Why is AvUC better than cross entropy within temperature scaling? (when used on top of SVI, or more generally). While the AvUC loss is motivated from Eqs.(1) and (2), it is actually defined only by Eqs.(3) and (4) which involves several choices: negative log, tanh, 1-tanh. Are these important for the success of the loss or would replacing tanh by some other monotonic function from all reals to [-1,+1] work also? --- After rebuttal --- Thanks for your response and proposed changes to the paper. Regarding my comments about MMCE [11] you have replied that 'Specifically, MMCE enables confidence calibration i.e. it accounts for the probability of the predicted class, but not overall model’s predictive uncertainty'. I agree with this criticism about MMCE. However, the ECE measure that you use as part of evaluation is also only taking into account the probability of the predicted class. Thus, I would have still liked to see comparisons against MMCE in the paper.


Review 4

Summary and Contributions: This paper proposes a differentiable version of the Accuracy v Utility Calibration (AUC) score that takes into account both predicted probabilities but also uncertainties. The differentiable score is then appended onto a standard SVI loss function as a secondary regularizer. Experiments are performed on deep image classification problems, where it is shown that accuracy remains similar but that predictive uncertainty is somewhat improved compared to simply ensembling. [Comments in brackets are updates post-rebuttal. After reading the authors' response, I've upgraded my score to a weak accept. Thank you for providing the clarifications, and I hope that you address the loss-calibrated inference methods in the camera ready.]

Strengths: Overall, I like the novelty and simplicity of the method as well as its empirical performance (with some qualms). Relevance: Bayesian deep learning and more generally uncertainty quantification in deep learning have become exceedingly popular recently - a NeurIPS tutorial last year and one this year at ICML, as practitioners increasingly want reliable predictive uncertainties for their deep network models. This paper resoundingly addresses the problem. Theoretical grounding: The differentiable approximation to the AvUC metric is pretty exciting, especially as it's only really possible (from the writing) if one has an estimate of the uncertainty at the new data point (e.g. something that comes from a Bayesian NN). Empirical evaluation: The experimental selection is good and extremely thorough throughout; I really appreciate the ImageNet experiment as it demonstrates that Bayesian (and probabilistic) methods are not just limited to small, older datasets. Novelty: Overall, the new loss function and the way that it's devised is clever, and pretty unexplored in the literature. It's a pretty neat use case of the SVI loss functions.

Weaknesses: I find that the main weaknesses are primarily on the empirical evaluation with a little bit on the theoretical side as well. I think these issues are straightforwardly fixable in the rebuttal. Empirical Evaluation: See detailed comments in the clarity section; the results, while pretty impressive on a first glance, could deal with a little more significance testing and a little less overselling. [Thanks for the clarifications, detailed comments in the next sections.] Significance: One potential weakness is that it's not placed well in the context of other works in the literature amazingly well. `I fear it could get sidelined because it's a bit of a "new loss" function paper rather than genre-breaking paper; however, the good execution and simplicity definitely help assuage my concerns here, so I don't have a real criticism in terms of significance. [Thanks for at least acknowledging this issue.] Novelty: I view this paper as a sort of spiritual successor to [3] in the deep learning literature and the older works [4,5] in the broader ML literature with a modern deep version published at ICLR last year [6] (no clear application to OoD or uncertainty quantification however). That's not necessarily a bad thing. Theoretical grounding: As this loss term term is a regularizer appended onto the top of an SVI loss term, it's not quite Bayesian inference any more -- it's really more of a loss-calibrated Bayesian method [1] applied to deep nets [2]. Citing these would certainly help to make the work more theoretically grounded and potentially easy to extend. As it is, the statistical / Bayesian motivation is somewhat nonexistent, which is somewhat concerning. References: [1] http://proceedings.mlr.press/v15/lacoste_julien11a/lacoste_julien11a.pdf [2] https://arxiv.org/pdf/1805.03901.pdf [3] http://proceedings.mlr.press/v80/kumar18a/kumar18a.pdf [4] http://www.icml-2011.org/papers/198_icmlpaper.pdf [5] https://icml.cc/Conferences/2004/proceedings/papers/132.pdf [6] https://arxiv.org/pdf/1908.10831.pdf

Correctness: Empirical methodology: There's two baselines around the loss term that I definitely see missing and would like to see in the rebuttal --- it's okay if they fail to train properly. 1) Would it be possible to train the NN solely through the AvUC loss -- that is in Eq. 5 throwing away the expected negative log likelihood? To my knowledge, this is slightly different than the ablation in Appendix E, which to my understanding just trains the threshold used to compute the metric. [Thanks for providing the experiment. It's interesting to see that the AvUC metric fails to converge when used only.] 2) Would it be possible to train a non-Bayesian NN to predict a probability and an uncertainty at each point while adding the AvUC loss term into the standard cross-entropy loss? This would be reminiscent of a heteroscedastic loss term for regression, and of the heteroscedastic classification loss term in [1] (which is now pretty foundational to the field of BDL). As a sidenote, the heteroscedastic classification loss term in [1] also would serve as a good OOD baseline in and of itself and probably ought to be compared to. [see 2) in add'tl feedback] These baselines are designed to parse out what the effect of the AvUC loss term is specifically and whether or not it really needs to require SVI in order to achieve good performance. Ablations: What is the effect of \beta =3 in terms of accuracy and uncertainty quantification? I see that it's set to \beta = 3 throughout the experiments in the supplemental materials, but there doesn't seem to be any understanding of how strong \beta needs to be to give demonstrable improvements. [See add't feedback] Are the results in the paper performed over different trials? I'm asking because many of the improvements are a bit small and could well be the results of randomness in the seed. [Thanks] References: [1] https://arxiv.org/abs/1703.04977

Clarity: Loss Clarity: Overall, the loss function is pretty well explained; although a toy example visualization would be very useful here. In particular, assessing the difference between how the weighting changes in comparison to a softmax loss would be nice, but this is a bonus point. Experimental Clarity: In some ways, while the empirical results as presented are quite flashy, it's rather tricky to evaluate and not particularly clear to interpret: 1) Consider using violin plots rather than boxplots to visualize the distributional shift in plots like Figure 1. As it is, it is rather difficult to see which method actually performs best as they all mostly perform comparably with a couple of exceptions. 2) Figure 1 caption: "SVI-AvUC consistently provides .... methods" feels a bit oversold given the box plots especially in comparison to ensembles (orange). Only at the very highest levels of corruption on ImageNet is it able to visually distinguish the 75% percentiles (presumably?) of the box plots between the methods. Consider instead reporting relative ranks of the methods and computing Spearman's rank correlation to report the results; it will also help you save space. [Thanks for the update, I'm now somewhat sure that the method is at least performing amongst the top two although the improvements seem small -- but not having run these types of experiments before, I don't know what strong improvment is.] 3) A similar concern to the above point can be made for Figure 2; here, it's probably not necessary to show all of these and just serves to make the important plots pointlessly small. Consider here just zooming in on the two or three most important and reporting the rest in a table. 4) Figure 3: All of these distributions look reasonably well separated; you may want to quantify the distances with either Jensen-Shannon (as in [1]) or Wasserstein. [Thanks.] 5) Table 1: Thanks for providing these -- are these for solely one run (ImageNet being 1 run is probably okay; the others not so much)? If so, the differences are small enough that it's not entirely clear if they are significant, although it is nice that SVI-AvUC does so well on those metrics (because it is optimized wrt them). Again, you probably want to use a rank correlation test here to assess relative ranks. [Thanks, you probably want to include the standard deviations as well.] 6) Thanks for the very detailed and thorough set of experiments throughout; these are very nice to investigate and to poke through. References: [1] https://arxiv.org/abs/1902.02476

Relation to Prior Work: In general, I think that the authors do a pretty good job of motivating themselves from prior work and motivating themselves from the Bayesian deep learning literature, as well as some of the works on more effectively measuring accuracy and calibration in deep learning. No obvious works clearly stand out to me as missing, while the method itself seems pretty novel in my reading.

Reproducibility: Yes

Additional Feedback: Line 400: typo, supplementary Line 385: two spaces before hendrycks and dietterich [From correctness 2): Thanks for the comparison to the non-Bayesian NN --- it seems like the additional regularization towards uncertainty is the primary mechanism of improvement. I do wonder what the loss/probability surface? of cross-entropy + AvUC looks like.] [From correctness/ablations: The lack of ablations around \beta is still a sticking point for me and does somewhat concern me.]

[Author Response · NeurIPS 2020]

We thank the reviewers for their thoughtful feedback, which will help us to improve the manuscript. We are encouraged that they
found our formulation and idea to be novel(**R1**,**R3**,**R4**), intuitive (**R1**,**R2**,**R4**), impactful(**R2**), clever(**R4**), well-motivated(**R4**), and
model agnostic(**R1**). We are glad they found empirical experiments to be detailed and comprehensive throughout with strong and
promising results(**R1**,**R2**,**R3**,**R4**). All reviewers found the paper is well-written and results are reproducible. We are gratified that **R4**
recognized the importance of AvUC loss *"differentiable approximation to the AvUC metric is pretty exciting"* and found that the
*paper resoundingly addresses the problem to get reliable predictive uncertainties*. We thank the reviewers for their kind words and
constructive feedback. We address reviewer comments below and will incorporate all feedback in the manuscript.

First, we would like to clarify that AvUC (accuracy vs uncertainty calibration) loss is devised to improve uncertainty calibration that
can be **combined** with existing losses without modifying the underlying principles (e.g. ELBO for Bayesian NN, cross-entropy for
non-Bayesian NN classifier). AvUC enables uncertainty calibration overcoming the challenge of unavailability of ground truth for
uncertainty. AvUC accounts for quality of principled aleatoric and epistemic uncertainties, which are important for many applications.

**R1:***Strong baseline:* Though ensemble provides higher accuracy and performs well under lower data shift intensities, we observe
SVI-AvUC is more robust under high data shift intensities and out-of-distribution settings (which is common in real-world) based
on the results evident from Table 1 and Fig 1,2,3. To demonstrate superiority of AvUC loss theoretically, for the camera ready
version we will include a discussion section as suggested and justify how AvUC serves as loss-calibrated inference to obtain reliable
uncertainties from approximate Bayesian inference (SVI) and discuss why AvUC is a proper uncertainty calibration loss in general.

*Application to regression problems:* Thank you for pointing us conformal prediction direction. We strongly believe our algorithm
can be extended to regression problems with modifications in the formulation of loss function leveraging the relationship between
prediction accuracy and uncertainty bounds. We are excited about this suggestion and we will seriously consider this for future work.

**R2:***Comparison to PAvPU[35]:* Though our work is motivated from PAvPU metric, PAvPU is not a differentiable function to be used
as a training loss and was originally proposed for uncertainty evaluation. We propose a trainable uncertainty calibration loss with
differentiable approximation to AvU metric as defined in Eqns 4.To the best of our knowledge, this is the first work on uncertainty
calibration that leverages the relationship between accuracy and uncertainty to train the model towards well-calibrated uncertainties.

**R2**, **R3**: *Theoretical justification*: Theoretically AvUC loss will be perfect 0 only when the model's uncertainty is perfectly calibrated
(AvU=1). In appendix D.1, we show how AvU score and AvUC loss are related to each other during training. As noted in Eqns
3 and 4, AvUC loss attempts to maximize AvU which will indirectly push the values of uncertainties up or down based on the
accuracy of predictions. When classification accuracy do not match uncertainty, $AvU \to 0$ and $\mathcal{L}_{AvUC} \to \infty$ forcing the gradient
computation exert towards $\mathcal{L}_{AvUC} \to 0$, which will happen when AvU score is pushed higher ($AvU \to 1$), enabling the model to
provide well-calibrated uncertainties. We will include a detailed section discussing why AvUC is a proper uncertainty calibration
loss and theoretically justifying how it serves as loss-calibrated inference method (as suggested by **R4**), in the manuscript.

**R3:***"Comparison with temperature scaling on top of SVI (SVI-TS):"* valid point; we have run experiments to compare with SVI-TS
and will include the results for this method in manuscript. Results from the experiment for ResNet50/ImageNet in below table.

| method | ECE↓ [data shift intensity (test / 1 / 2 / 3 / 4 / 5)] | UCE↓ [data shift intensity (test / 1 / 2 / 3 / 4 / 5)] |
|---|---|---|
| SVI-TS | 0.024 / 0.030 / 0.048 / 0.072 / 0.098 / 0.123 | 0.117 / 0.164 / 0.199 / 0.235 / 0.269 / 0.294 |
| SVI-AvUTS | **0.019 / 0.027 / 0.029 / 0.041 / 0.059 / 0.080** | **0.088 / 0.144 / 0.142 / 0.173 / 0.203 / 0.226** |

*"Comparisons have been made only on top of SVI"*: In appendix D.6, comparison of AvUTS on top of vanilla is presented. Also
based on request from **R4**, we provide results of AvUC results applied to non-Bayesian NN here below (line 49-50). In the final
version, we will include the evaluations on top of both SVI and Vanilla baselines (Bayesian and non-Bayesian method). In L194-198
of main manuscript, we mention why we chose SVI as a baseline to evaluate our methods.

*Whether AvUC is a proper loss; Comparison to existing losses including MMCE[11] and cross entropy within temperature scaling*:
In appendix D.1, we show how AvUC loss converges based on uncertainty calibration (AvU score). When compared to existing
losses including MMCE, AvUC loss accounts for quality of principled aleatoric and epistemic uncertainties while training the model,
improves uncertainty calibration, can be combined with existing losses for training or used solely for post-hoc calibration and
overcomes the challenge of unavailability of ground truth uncertainty. Specifically, MMCE enables confidence calibration i.e. it
accounts for the probability of the predicted class, but not overall model's predictive uncertainty.

**R4:** Thanks for pointing the references on loss-calibrated inference methods, we agree including these citations will help to make
the work more theoretically grounded and extendable as suggested. We will incorporate the feedback in final version. Regarding
*training the NN solely through the AvUC loss*, we were not able to achieve training convergence in this case as AvUC is devised as an
uncertainty calibration loss. However, we are able to perform temperature scaling solely with AvUC loss (AvUTS method).

*Would it be possible to train a non-Bayesian NN:* We trained vanilla baseline (ResNet20/CIFAR10) with AvUC loss in addition to
cross-entropy as requested, the results are better (lower ECE and UCE) than solely training with cross-entropy. Also, in appendix
D.6 we have presented the results of AvUTS applied to non-Bayesian NN (ResNet50/ImageNet).

| method | ECE↓ [data shift intensity (test / 1 / 2 / 3 / 4 / 5)] | UCE↓ [data shift intensity (test / 1 / 2 / 3 / 4 / 5)] |
|---|---|---|
| Vanilla | 0.046 / 0.098 / 0.139 / 0.183 / 0.236 / 0.315 | 0.038 / 0.085 / 0.122 / 0.162 / 0.212 / 0.285 |
| Vanilla-AvUC | **0.034 / 0.074 / 0.102 / 0.138 / 0.185 / 0.250** | **0.017 / 0.044 / 0.064 / 0.092 / 0.134 / 0.189** |

*Are the results in the paper performed over different trials?:* Yes, we used five different trails for CIFAR10 experiments (In Fig 2 and
F9, shading in the plots shows the standard errors from different trails). For ImageNet experiments, results are reported over one trial.

*Results presentation:* Thank you for suggesting the presentation improvements of reported results, we will address these in camera
ready version. We observed SVI-AvUC results show better spearman's $\rho$ compared to other methods indicating least correlation
of calibration errors w.r.t dataset shift [ImageNet/ECE - Vanilla:1.0, Ensemble: 0.82, SVI-AvUTS: 0.94, SVI-AvUC: **0.2**]. We
will report relative ranks and include spearman rank correlation coefficient covering all the baselines, metrics and datasets in the
manuscript as suggested. We have quantified the distribution separation of uncertainties in Fig 3 with Wasserstein distance [Vanilla:
2.73, Temp scaling: 2.90, Ensemble: 3.02, Dropout: 3.38, SVI: 3.73, SVI-AvUTS: 3.94, SVI-AvUC: **4.29**].

[Meta-Review · NeurIPS 2020]

The paper proposes a novel loss called accuracy versus uncertainty calibration (AvUC) for improving uncertainty estimation in deep learning. The initial scores were borderline and reviewers raised a few concerns. I thank the authors for writing a thoughtful rebuttal with additional experiments that were appreciated by the reviewers. During the discussion, all the reviewers agreed that the author rebuttal addresses the major concerns and several of them increased scores as well. I have read the paper carefully as well and I recommend accept due to the following reasons: Pros: - Simplicity of the approach - Well-written paper - Extensive experimental results including calibration under dataset shift, OOD detection showing the benefits of the proposed approach - Accompanying code which should make it easier for folks in the community to build on this work Suggestions for camera ready: - Theoretical justification: the points in the rebuttal should be moved to the main text (especially connections to loss calibrated inference and justification as as a proper loss) - Experimental results: Some of the additional ablations requested were added during rebuttal, please add them to the main text as this strengthens the paper - There's a couple of other minor comments raised by the reviewers which weren't completely addressed in the rebuttal. But these are relatively minor and I encourage the authors to address them as well in the camera-ready.